# Spectrally-Guided Diffusion Noise Schedules

**Carlos Esteves** [1]   **Ameesh Makadia** [1]

## Abstract

Denoising diffusion models are widely used for high-quality image and video generation. Their performance depends on noise schedules, which define the distribution of noise levels applied during training and the sequence of noise levels traversed during sampling. Noise schedules are typically handcrafted and require manual tuning across different resolutions. In this work, we propose a principled way to design per-instance noise schedules for pixel diffusion, based on the image's spectral properties. By deriving theoretical bounds on the efficacy of minimum and maximum noise levels, we design "tight" noise schedules that eliminate redundant steps. During inference, we propose to conditionally sample such noise schedules. Experiments show that our noise schedules improve generative quality of single-stage pixel diffusion models, particularly in the low-step regime.

## 1. Introduction

Denoising diffusion models (Sohl-Dickstein et al., 2015; Ho et al., 2020) are generative models based on learning to reverse a noising process that progressively destroys the data. They are the foundation of state-of-the-art media generation since Latent Diffusion Models (LDM) (Rombach et al., 2022), which operate on the latent space of a visual autoencoder. This combination produced a series of popular applications in image (Ramesh et al., 2022; Podell et al., 2024) and video generation (Blattmann et al., 2023; Brooks et al., 2024; DeepMind, 2025).

Despite the LDM dominance, they have disadvantages – the generation quality is inherently capped by the autoencoder/tokenizer quality, and the two-stage training can be cumbersome, since there is no clear connection between the autoencoder reconstruction and generative performance (Yu et al., 2024; Hansen-Estruch et al., 2025). Some alternatives avoid generation on latent space but still require multi-stage training for upsampling in pixel space (Nichol & Dhariwal, 2021; Ho et al., 2022; Saharia et al., 2022).

These disadvantages motivated recent renovations in single-stage pixel diffusion (Hoogeboom et al., 2023; 2025; Chen et al., 2025; Wang et al., 2025; Li & He, 2025; Yu et al., 2025), with improvements in model architecture and training protocol reducing the gap to LDMs. While significant progress has been made, LDMs still show better generative quality at lower computational cost. One of the reasons is that LDMs require up to one order of magnitude fewer denoising steps than pixel diffusion (Hoogeboom et al., 2025).

The noise level of each denoising step is determined by the *noise schedule*, which is typically handcrafted as a linear or cosine-like curve increasing with the time step $t$. Recent approaches such as Simple Diffusion (Hoogeboom et al., 2023) adapt the schedule across resolutions by shifting the curve. As illustrated in Fig. 2 (left), these heuristics relate to the power spectrum observed in natural images – higher resolution images have more energy at lower frequencies, thus more noise is needed to destroy the signal. Since following these dataset-level spectral trends with heuristics has been successful, we posit that adapting the schedule to the spectrum of each instance can provide further improvements.

In this work, we observe that typical noise schedules are inefficient, prescribing inappropriate noise levels for a significant number of steps (see Fig. 1). We design a principled noise schedule that adapts to each image based on its spectral properties, and show that it improves quality with significantly less deterioration under reduced number of denoising steps.

Our contributions are as follows. 1) We design "tight" per-instance noise schedules that follow the signal's power spectrum. 2) We derive theoretical bounds on the efficacy of minimum and maximum noise levels. 3) We propose a conditional mechanism to predict the power spectrum and corresponding noise schedule prior to image sampling.

We demonstrate that our schedules improve generative quality compared to baseline pixel diffusion models, with particularly large margins in the low-step regime.

---

[1]Google Research. Correspondence to: Carlos Esteves <machc@google.com>.

*Proceedings of the 43ʳᵈ International Conference on Machine Learning*, Seoul, South Korea. PMLR 306, 2026. Copyright 2026 by the author(s).

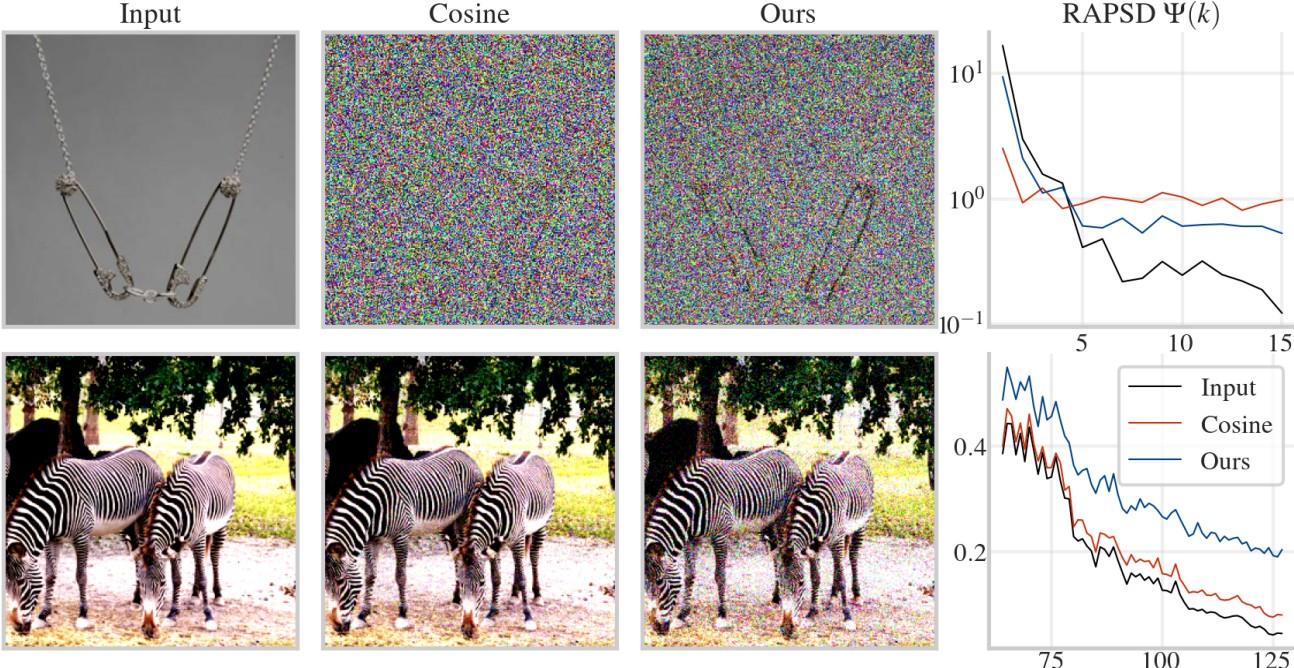

*Figure 1.* Our "tight" schedules adapt to each instance's spectrum, ensuring effective noise levels at all steps. *Top:* An image with low energy on low frequencies. The standard cosine noise schedule destroys the signal at $t = 0.5$, which means that at least half of the training steps would apply excessive noise for this input. Our adaptive schedule preserves the low frequency content – notice that the object outline is still visible. *Bottom:* An image with high energy on high frequencies. The cosine schedule barely changes the input at $t = 0.1$ – notice that the RAPSD curves between the cosine schedule and the input are close and correlated. This means that at least $10\%$ of the training steps would apply insufficient noise. Our schedule is effective at destroying a part of the high-frequency content at this level.

## 2. Related work

**Diffusion models** were introduced by Sohl-Dickstein et al. (2015) and received increased attention in media generation since DDPM (Ho et al., 2020). Rombach et al. (2022) laid out the core ideas for current state-of-the-art LDMs. Here we depart from LDMs and adopt pixel diffusion, closely following the formulation of VDM++ (Kingma et al., 2021; Kingma & Gao, 2023) and the architectures and protocols of Simpler Diffusion (Hoogeboom et al., 2023; 2025).

**The noise schedule** is a crucial component of diffusion models and determines the noise level during training and sampling. Ho et al. (2020) adopted $x_t = \sqrt{1 - \beta_t}x_{t-1} + \beta_t\epsilon$, where $\beta_t$ increased linearly with the time step and $\epsilon \sim \mathcal{N}(0, I)$. Nichol & Dhariwal (2021) introduced the widely used cosine schedule, $x_t = \sqrt{\alpha_t}x_0 + \sqrt{1 - \alpha_t}\epsilon$, where $\alpha_t$ decays little near $t = 0$ and $t = 1$ and linearly in the middle. EDM (Karras et al., 2022) established a log-normal distribution of noise levels to prioritize intermediate levels. Hang et al. (2025) connected the noise schedules with importance sampling of logSNR and verified the importance of intermediate levels (zero logSNR). Lin et al. (2024) corrected a number of flaws identified in diffusion implementations. Esser et al. (2024) extended the Kingma &

Gao (2023) analysis to rectified flows (Liu et al., 2022; Albergo & Vanden-Eijnden, 2023; Lipman et al., 2023) and again found it best to prioritize intermediate noise levels. These methods prescribe a global noise schedule, while our method prescribes a different schedule for each instance.

**Noise schedules across resolutions** Jabri et al. (2023) introduced the sigmoid schedule with temperature that downweights extreme noise levels; they also noticed that increasing the temperature shifts the schedule towards more noise and performs better at higher resolution. Chen (2023) suggested a simple idea of scaling inputs by a constant factor to adjust the noise factor so that more noise can be introduced for higher resolutions. Hoogeboom et al. (2023) observed directly that more noise is needed to destroy high-resolution signals, and proposed to shift the noise schedule according to the input resolution. They further proposed a timestep dependent shift that happens at low signal-to-noise ratio (SNR) and not high. These design decisions relate to the power spectrum trends depicted in Fig. 2 (left); the power at lower frequencies increases with the image resolution, which also introduces new low-powered high frequencies, justifying the timestep-dependent shifts. In this work, we explicitly use each instance's power spectrum to determine its noise schedule; in aggregate this gives rise to similar

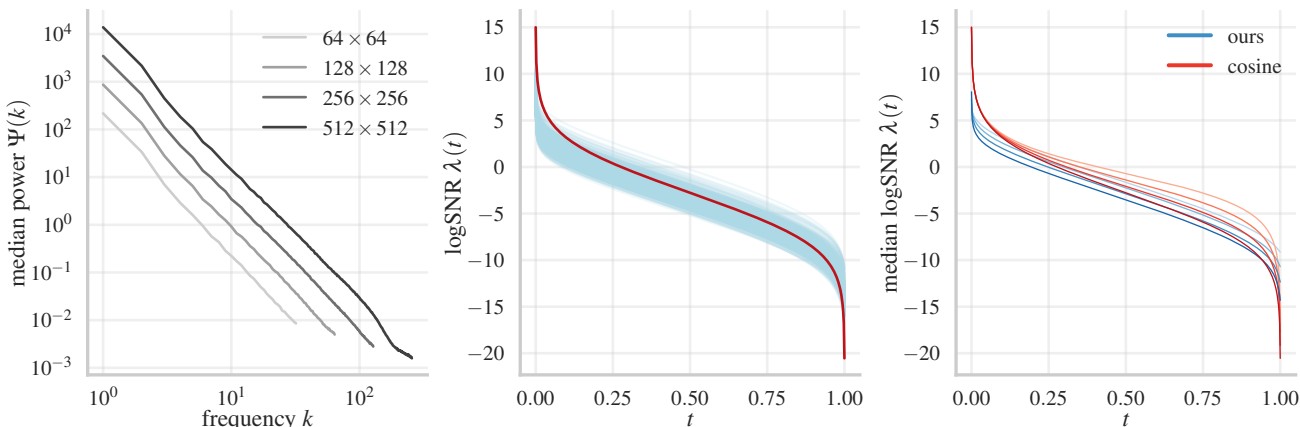

*Figure 2.* Our noise schedules vary per instance based on its spectral properties. *Left:* Median power per frequency for ImageNet at multiple resolutions (increasing from light to dark). The power spectrum of natural images follows a power law whose trends explain current noise schedule tuning heuristics. We eschew such heuristics and use each instance's spectrum to determine its schedule. *Middle:* Cosine schedule and ours for 1000 images from ImageNet $256 \times 256$. *Right:* Median noise schedules for the same set of images, at $128 \times 128$, $256 \times 256$, and $512 \times 512$ (light to dark color). Our schedules avoid excessively high and low noise values, while following similar trends to the baseline across resolutions without any hyperparameter change.

trends as prior work (see Fig. 2), but our schedules naturally adapt to each instance and resolution without handcrafting.

**Learning noise schedules** Kingma et al. (2021) showed that, in theory, the noise schedule does not matter since the loss reduces to an integral between minimum and maximum SNR. Kingma & Gao (2023) observed that, in practice, the schedule affects the variance of the Monte-Carlo estimation of the loss which in turn affects optimization efficiency. They proposed an adaptive noise schedule based on the training loss, which resulted similar quality but potentially faster training. Sahoo et al. (2024) learned a per-pixel polynomial noise schedule that optimizes a tighter evidence lower bound (ELBO), with a focus on improving density estimation. Our method is simpler, connecting the noise schedule to each instance's spectral properties, while showing clear improvements in quality and denoising steps reduction.

**Spectral analysis and diffusion models** There is a growing literature in understanding diffusion through the lens of spectral analysis. Rissanen et al. (2023) and Dieleman (2024) connected diffusion with spectral autoregression, as both processes generate images by progressively introducing frequencies. Falck et al. (2025) showed evidence against this interpretation and introduced EqualSNR to enforce that all frequencies are corrupted equally during the forward process, achieving similar quality. Huang et al. (2024) found improvements by using blue instead of white noise. Jiralerspong et al. (2025) showed improvements by designing colored noise with more power on low-frequencies than high. Based on spectral analysis of the denoising process of Gaussian-generated data with arbitrary covariance, Benita et al. (2025) optimized a noise schedule end-to-end for

a given dataset, resolution, and number of sampling steps. The optimized schedules followed similar trends to the hand-crafted cosine schedule. These references still prescribed noise schedules for a whole dataset, while we propose a per-instance strategy that adapts to the spectral diversity within the dataset.

**Reducing the number of denoising steps** Diffusion modeling in latent space (Rombach et al., 2022) naturally requires fewer denoising steps than in higher dimensional pixel space. Distillation is a popular strategy for step count reduction (Song et al., 2023; Salimans & Ho, 2022; Yin et al., 2024; Nguyen & Tran, 2024; Meng et al., 2023; Salimans et al., 2024). Another class of techniques are rectified flows (Liu et al., 2022; Albergo & Vanden-Eijnden, 2023; Lipman et al., 2023; Lee et al., 2024), and mean flows (Geng et al., 2025). These are complementary to our per-instance noise schedules, and could potentially be combined.

## 3. Background

The forward time diffusion process is given by

$$x_t = \alpha_t x_0 + \sigma_t \epsilon, \quad \epsilon \sim \mathcal{N}(0, I), \quad 0 \leq t \leq 1, \quad (1)$$

where $x_0$ is a clean image. The noise schedule determines $\alpha_t$ and $\sigma_t$; for example, $\alpha_t = \cos(\pi t/2)$ defines a cosine schedule. Schedules are often defined in terms of the logSNR $\lambda(t) = \log(\alpha_t^2/\sigma_t^2)$ (Kingma et al., 2021).

Training minimizes the sigmoid-weighted ELBO, following Kingma & Gao (2023) and Hoogeboom et al. (2025).

$$\mathcal{L}_\theta(x_0; t, y) = -\lambda'(t)e^b\boldsymbol{\sigma}(\lambda(t)-b)\|x_\theta(x_t; c)-x_0\|_2^2, \quad (2)$$

**Algorithm 1** Training with Spectrally-Guided Schedules

**Input:** Dataset $\mathcal{D}$, model $x_\theta$
**for** $i = 1$ **to** num_training_steps **do**
    Sample data $x_0 \sim \mathcal{D}$ with label/prompt $y$
    Sample time $t \sim \mathcal{U}(0, 1)$ and noise $\epsilon \sim \mathcal{N}(0, I)$
    Compute RAPSD $\Psi_{x_0}$ from $x_0$     ▷ Eqs. (6) and (7)
    Fit $\tilde{\Psi}_{x_0}(k) = \beta k^\alpha$ to $\Psi_{x_0}$     ▷ Section 4.4
    Compute schedule $\lambda_M$ using $\tilde{\Psi}_{x_0}$ ▷ Eqs. (19) to (23)
    Compute $\alpha_t, \sigma_t$, and $x_t$ using $\lambda_M$ ▷ Eqs. (1) and (17)
    Update params $\theta$ given $\nabla_\theta \mathcal{L}$ over a batch     ▷ Eq. (2)
**end for**

**Algorithm 2** Spectrally-Guided Sampling

**Input:** Label/prompt $y$, number of denoising steps $N$.
Sample $\alpha, \beta$ from RAPSD sampler     ▷ Eqs. (24) to (26)
Define spectrum $\tilde{\Psi}_x(k) = \beta k^\alpha$
Compute schedule $\lambda_M$ using $\tilde{\Psi}_x$     ▷ Eqs. (19) to (23)
Sample $x_1 \sim \mathcal{N}(0, I)$
**for** $i = N$ **to** $1$ **do**
    $t \leftarrow i/N, \quad s \leftarrow (i-1)/N$
    Compute $\alpha_t, \sigma_t, \alpha_s, \sigma_s$ from $\lambda_M$     ▷ Eq. (17)
    Get $x_s$ from $x_t$ given $\epsilon \sim \mathcal{N}(0, I)$     ▷ Eqs. (3) to (5)
**end for**
**Return** $x_0$

where $t \sim \mathcal{U}(0, 1)$, $b$ is a constant bias, $\boldsymbol{\sigma}$ is the sigmoid function, and $x_\theta$ is a neural network. A typical conditioning is $c = (t, y)$, where $y$ is the class label or text prompt. After training, we use ancestral sampling for generation,

$$\hat{x}_\theta = x_\theta(x_t; c) + w(x_\theta(x_t; c) - x_\theta(x_t; c_\emptyset)), \quad (3)$$

$$x_s = \alpha_s \hat{x}_\theta + \frac{\alpha_t \sigma_s^2}{\alpha_s \sigma_t^2}(x_t - \alpha_t \hat{x}_\theta) + \sigma_{t \to s} \epsilon, \quad (4)$$

$$\sigma_{t \to s} = \sigma_s^{1-\gamma} \sigma_t^\gamma \sqrt{1 - \exp(\lambda(t) - \lambda(s))}, \quad (5)$$

where $w$ is the scale of classifier-free guidance (Ho & Salimans, 2021), $c_\emptyset$ is the conditioning with a null label/prompt embedding, $s < t$ and $\gamma$ is a hyperparameter. This process starts from pure noise and is repeated until $s = 0$.

## 4. Method

### 4.1. Preliminaries

Consider a discrete signal $x : \{0, \dots, N-1\}^d \to \mathbb{R}$. Its Discrete Fourier Transform (DFT) is,

$$\hat{x}(u) = \frac{1}{N^{d/2}} \sum_n x(n) \exp\left(-i\frac{2\pi}{N}u^\top n\right). \quad (6)$$

The power spectral density is $P_x(u) = |\hat{x}(u)|^2$. The radially-averaged power spectral density (RAPSD) is

$$\Psi_x(k) = \frac{1}{N_k} \sum_{u:\text{round}(\|u\|_2)=k} P_x(u), \quad (7)$$

where $k = \|u\|_2$ is the scalar frequency, and $N_k$ the number of frequency vectors $u$ that satisfy the rounding. In this work, we focus on RGB images so $\Psi_x(k) = \frac{1}{3}\sum_{c=1}^3 \Psi_{x_c}(k)$, where $c$ indexes the color channels, and $0 \le k \le N_f$, with $N_f$ being the Nyquist frequency (half of the image side).

For natural images, ignoring the DC component $u = \mathbf{0}$ so $k \ge 1$, the RAPSD typically follows a power law,

$$\Psi_x(k) \approx k^\alpha \beta, \quad (8)$$

where $\alpha < 0$ and $\beta > 0$. Furthermore, $\alpha \approx -2$ and for range $[-1, 1]$, $\Psi_x(1) \gg 1$ and $\Psi_x(N_f) \ll 1$ (Field, 1987; van der Schaaf & van Hateren, 1996; Torralba & Oliva, 2003). The RAPSD of white noise is one everywhere.

### 4.2. Noise level per frequency

Our main contribution is a per-instance noise schedule that follows the power spectrum, avoiding too much or too little noise. This amounts to prescribing 1) the minimum amount of noise that destroys the signal, 2) the maximum amount of noise that preserves the signal, and 3) everything in between.

At some noise level $q$, we obtain the expected RAPSD of the noised input $z_q = \alpha_q x_0 + \sigma_q \epsilon$ as (see proof in Section A),

$$\Psi_{z_q}(k) = \alpha_q^2 \Psi_{x_0}(k) + \sigma_q^2. \quad (9)$$

Suppose we set the noise level $\sigma_q$ proportional to the power at some frequency $q$, with $\kappa_q > 0$,

$$\sigma_q^2 = \kappa_q \alpha_q^2 \Psi_{x_0}(q). \quad (10)$$

Assuming a variance-preserving schedule, $\alpha_q^2 + \sigma_q^2 = 1$, we substitute Eq. (10) to solve for $\alpha_q^2$ and $\sigma_q^2$:

$$\alpha_q^2 + \kappa_q \alpha_q^2 \Psi_{x_0}(q) = 1 \implies \alpha_q^2 = \frac{1}{1 + \kappa_q \Psi_{x_0}(q)}, \quad (11)$$

$$\sigma_q^2 = \kappa_q \alpha_q^2 \Psi_{x_0}(q) = \frac{\kappa_q \Psi_{x_0}(q)}{1 + \kappa_q \Psi_{x_0}(q)}. \quad (12)$$

Substituting $\alpha_q^2$ and $\sigma_q^2$ back into Eq. (9), we obtain the expected RAPSD for all $k$:

$$\Psi_{z_q}(k) = \left(\frac{1}{1 + \kappa_q \Psi_{x_0}(q)}\right) \Psi_{x_0}(k) + \frac{\kappa_q \Psi_{x_0}(q)}{1 + \kappa_q \Psi_{x_0}(q)}$$
$$= \frac{\Psi_{x_0}(k) + \kappa_q \Psi_{x_0}(q)}{1 + \kappa_q \Psi_{x_0}(q)}. \quad (13)$$

*Table 1.* Class-conditional generation on ImageNet. We compare our spectral noise scheduling against recent single-stage pixel diffusion baselines. The fairest comparison is against SiD2 (Hoogeboom et al., 2025); we use exactly the same architecture and training protocol except for our contributions described in Section 4. We outperform the baselines in most metrics, while needing fewer denoising steps than SiD2. We reproduce SiD2 it and compute additional metrics besides the reported FID; originally reported results are quoted next to reproduced. Section D.3 describes each metric. Ours and SiD2 values are averaged over 5 sets of generations with different seeds. **NFE**: number of function evaluations (denoising steps). **Adapt.**: PixelFlow uses a (slower) solver with adaptive number of steps.

| MODEL | PARAMS | NFE | FID ↓ | sFID ↓ | IS ↑ | PRECISION ↑ | RECALL ↑ |
|---|---|---|---|---|---|---|---|
| **IMAGENET 512 × 512** | | | | | | | |
| JIT-G (LI & HE, 2025) | 2B | - | 1.78 | - | 306.8 | - | - |
| PIXNERD-XL/16 (WANG ET AL., 2025) | 700M | 100 | 2.84 | 5.95 | 245.6 | **0.80** | 0.59 |
| SID2, SMALL | 397M | 512 | 2.19 (2.19) | 4.30 | 295.3 | 0.72 | 0.63 |
| **OURS, SMALL** | 399M | 256 | 1.79 | 4.39 | 306.1 | 0.73 | **0.64** |
| SID2, FLOP HEAVY | 397M | 512 | 1.53 (1.48) | 3.98 | 306.2 | 0.74 | 0.63 |
| **OURS, FLOP HEAVY** | 399M | 320 | **1.45** | **3.91** | **310.0** | 0.74 | 0.63 |
| **IMAGENET 256 × 256** | | | | | | | |
| JIT-G (LI & HE, 2025) | 2B | - | 1.82 | - | 292.6 | 0.79 | 0.62 |
| PIXELFLOW-XL (CHEN ET AL., 2025) | 677M | ADAPT. | 1.98 | 5.83 | 282.1 | **0.81** | 0.60 |
| PIXNERD-XL/16 (WANG ET AL., 2025) | 700M | 100 | 2.15 | 4.55 | **297.0** | 0.79 | 0.59 |
| PIXELDIT-XL (YU ET AL., 2025) | 797M | 100 | 1.61 | 4.68 | 292.7 | 0.78 | 0.64 |
| SID2, SMALL | 397M | 512 | 1.68 (1.72) | 4.04 | 288.2 | 0.72 | **0.65** |
| **OURS, SMALL** | 399M | 256 | 1.42 | 3.82 | **297.0** | 0.73 | **0.65** |
| SID2, FLOP HEAVY | 397M | 512 | 1.37 (1.38) | 3.83 | 286.3 | 0.73 | **0.65** |
| **OURS, FLOP HEAVY** | 399M | 256 | **1.32** | **3.71** | 294.2 | 0.74 | 0.64 |
| **IMAGENET 128 × 128** | | | | | | | |
| SID2, SMALL | 397M | 512 | 1.62 | 3.76 | 220.0 | 0.73 | 0.64 |
| **OURS, SMALL** | 399M | 160 | 1.43 | 3.65 | **223.9** | **0.74** | 0.64 |
| SID2, FLOP HEAVY | 393M | 512 | **1.30** (1.26) | 3.64 | 210.9 | 0.73 | **0.65** |
| **OURS, FLOP HEAVY** | 395M | 160 | 1.30 | **3.53** | 204.8 | 0.73 | 0.64 |

**Largest noise** For $q = 1$ we have $\Psi_{x_0}(q) \gg 1$. Thus,

$$\Psi_{z_1}(k) \approx \frac{\Psi_{x_0}(k) + \kappa_1 \Psi_{x_0}(1)}{\kappa_1 \Psi_{x_0}(1)} = 1 + \frac{\Psi_{x_0}(k)}{\kappa_1 \Psi_{x_0}(1)}. \quad (14)$$

For $k = 1$, we have $\Psi_{z_1}(1) \approx 1 + {}^1\!/\kappa_1$, while for $k > 1$ we have $\Psi_{x_0}(k) \lesssim \Psi_{x_0}(1)$ and $\Psi_{z_1}(k) \lesssim 1 + {}^1\!/\kappa_1$.[1] Since $\Psi_{z_1}(k) \lesssim 1 + {}^1\!/\kappa_1$ holds for all $k$, the greater $\kappa_1$ is, the closer the RAPSD is to that of unit Gaussian noise. This measures how close the signal is to pure noise, and determines our maximum noise level. We define $\kappa_{\max} = \kappa_1$.

**Smallest noise** For $q = N_f$, $\Psi_{x_0}(q) \ll 1$. From Eq. (13),

$$\Psi_{z_{N_f}}(k) \approx \Psi_{x_0}(k) + \kappa_{N_f} \Psi_{x_0}(N_f). \quad (15)$$

Now for $k = N_f$, we have ${}^{\Psi_{z_{N_f}}(N_f)}\!/_{\Psi_{x_0}(N_f)} \approx 1 + \kappa_{N_f}$, while for $k < N_f$ we have $\Psi_{x_0}(k) \gtrsim \Psi_{x_0}(N_f)$ and ${}^{\Psi_{z_{N_f}}(k)}\!/_{\Psi_{x_0}(k)} \lesssim 1 + \kappa_{N_f}$. Thus, the smaller $\kappa_{N_f}$ is, the closer the RAPSD of $z_{N_f}$ is to that of $x$; this determines our minimum noise level. We define $\kappa_{\min} = \kappa_{N_f}$.

**In-between noise** Since there are several orders of magnitude between $\kappa_{\max}$ and $\kappa_{\min}$ (for example, for 1% tolerance

---

[1] Typically $\Psi_{x_0}$ is monotonically decreasing so $\Psi_{x_0}(k) < \Psi_{x_0}(1)$ and $\Psi_{z_1}(k) < 1 + {}^1\!/\kappa_1$. We ensure this in Section 4.4.

we have $\kappa_{\max} = 100$ and $\kappa_{\min} = 0.01$), we prescribe the noise level at any frequency $q$ by interpolating in log-space,

$$\kappa_q = \kappa_{\max}^{\frac{N_f - q}{N_f - 1}} \kappa_{\min}^{\frac{q - 1}{N_f - 1}}. \quad (16)$$

### 4.3. Noise schedule

Last section showed appropriate noise levels for each discrete frequency of the signal. Because diffusion models require the noise level to strictly increase over time, we find a continuous, monotonically decreasing noise schedule in terms of $\lambda(t) = \log \alpha_t^2/\sigma_t^2$, with $t \in [0, 1]$. Under the variance-preserving assumption,

$$\alpha_t = \sqrt{\boldsymbol{\sigma}(\lambda(t))}, \quad \sigma_t = \sqrt{\boldsymbol{\sigma}(-\lambda(t))}. \quad (17)$$

We define $\tilde{\Psi}_{x_0} : \mathbb{R} \to \mathbb{R}$ as a monotonic continuous approximation of $\Psi_{x_0}$, and $\kappa_t = \kappa_{\max}^t \kappa_{\min}^{1-t}$. We propose three simple heuristics to map between $t \in [0, 1]$ and $q \in [1, N_f]$.

**Frequency-focused schedule** The simplest such map is

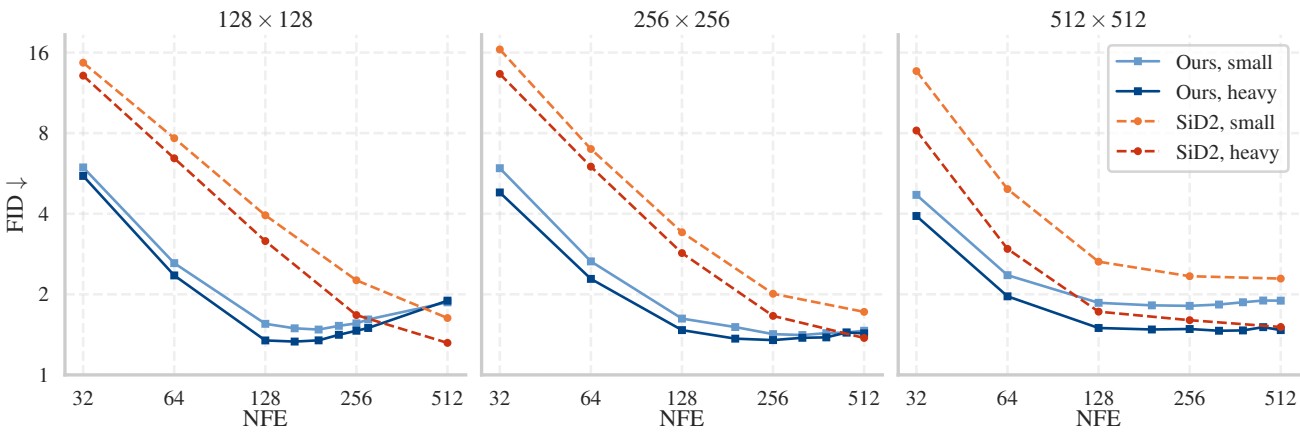

*Figure 3.* Comparison against the SiD2 (Hoogeboom et al., 2025) baseline on ImageNet, at different number of function evaluations (NFE), or denoising steps. Our model outperforms the baseline at the optimal number of steps, and the gap widens as the number of steps reduces. Interestingly, our "tight" schedules exhibit a slight FID worsening at high number of steps, we investigate this further on Section C.1.

linear, $\mu_F(t) = N_f + (1 - N_f)t$, which yields the schedule,

$$\sigma_t^2 = \kappa_t \alpha_t^2 \tilde{\Psi}_{x_0}(\mu_F(t)), \qquad (18)$$

$$\lambda_F(t; x_0) = -\log \kappa_t - \log \tilde{\Psi}_{x_0}(\mu_F(t)), \qquad (19)$$

where Eq. (19) comes from computing the logSNR in Eq. (18). We denote $\lambda_F$ the *frequency-focused* schedule, because, since $t$ is sampled uniformly, the noise corresponding to each frequency appears at the same rate. Since most of the frequencies have low power, noise levels are low, focusing more on image details than on its coarse structure.

**Power-focused schedule** We propose an alternative map, where we use $\tilde{\Psi}_{x_0}$ as a probability distribution function (PDF). Since power is concentrated in lower frequencies, this will cover higher noise levels more often, focusing more on coarse image structure than on high frequency details.

Let $Z = \int_1^{N_f} \tilde{\Psi}_{x_0}(u)du$ be the normalization constant. We define the cumulative distribution function (CDF) $F_{x_0}$ and the *power-focused* schedule $\lambda_P$ as follows,

$$F_{x_0}(q) = \frac{1}{Z} \int_1^q \tilde{\Psi}_{x_0}(u)du, \qquad (20)$$

$$\mu_P(t; x_0) = F_{x_0}^{-1}(1 - t), \qquad (21)$$

$$\lambda_P(t; x_0) = -\log \kappa_t - \log \tilde{\Psi}_{x_0}(\mu_P(t; x_0)). \qquad (22)$$

**Mixed schedule** To generate high quality images, the model needs to balance its focus on coarse structure and high-frequency details. Prior work suggested sampling intermediate noise levels more often to achieve this (Nichol & Dhariwal, 2021; Karras et al., 2022; Hang et al., 2025). We found that combining the frequency- and power-focused schedules effectively balances the coarse and fine focus and results in the best performance. We define the *mixed*

schedule $\lambda_M$ as, simply,

$$\lambda_M(t; x_0) = \frac{1}{2}(\lambda_F(t; x_0) + \lambda_P(t; x_0)). \qquad (23)$$

Figure 2 shows this schedule, compared with the SiD (Hoogeboom et al., 2023; 2025) shifted-cosine baselines. Our mixed schedule exhibits similar prioritization of intermediate levels, but tighter bounds. It maintains consistent trends across resolutions without per-resolution tuning, unlike the SiD baselines. Using the mixed schedule is a heuristic design choice, but its per-instance bounds follow the theoretical justification from Section 4.2, and it requires tuning of only two hyperparameters ($\kappa_{\min}$, $\kappa_{\max}$) that are kept constant across different datasets and resolutions.

### 4.4. Fitting and sampling the power spectrum

The schedules defined in Section 4.3 come from each image during training time, by computing its RAPSD. However, they are not available during sampling, when the model generates an image given only the conditioning. Our solution is to sample the RAPSD before generating the image.

We approximate the RAPSD as a power-law $\tilde{\Psi}_{x_0}(k) = k^\alpha \beta$ following Eq. (8) to 1) reduce the number of parameters to only two ($\alpha$ and $\beta$), and 2) ensure monotonicity. This is computed with least-squares in log-space and enables finding closed-form equations for our schedules, see Section B.

We train an RAPSD sampler $S$ that maps $y$ (class label or text prompt) to the parameters of a Gaussian Mixture Model (GMM) of $C$ components: weights $w_c$, 2D means $\mu_c$ and 2D diagonal covariances $\sigma_c$. The GMM represents the distribution of $v_1 = \log \tilde{\Psi}_x(1)$ and $v_2 = \log \tilde{\Psi}_x(N_f)$.

For class-conditional generation the map $S$ is a simple linear

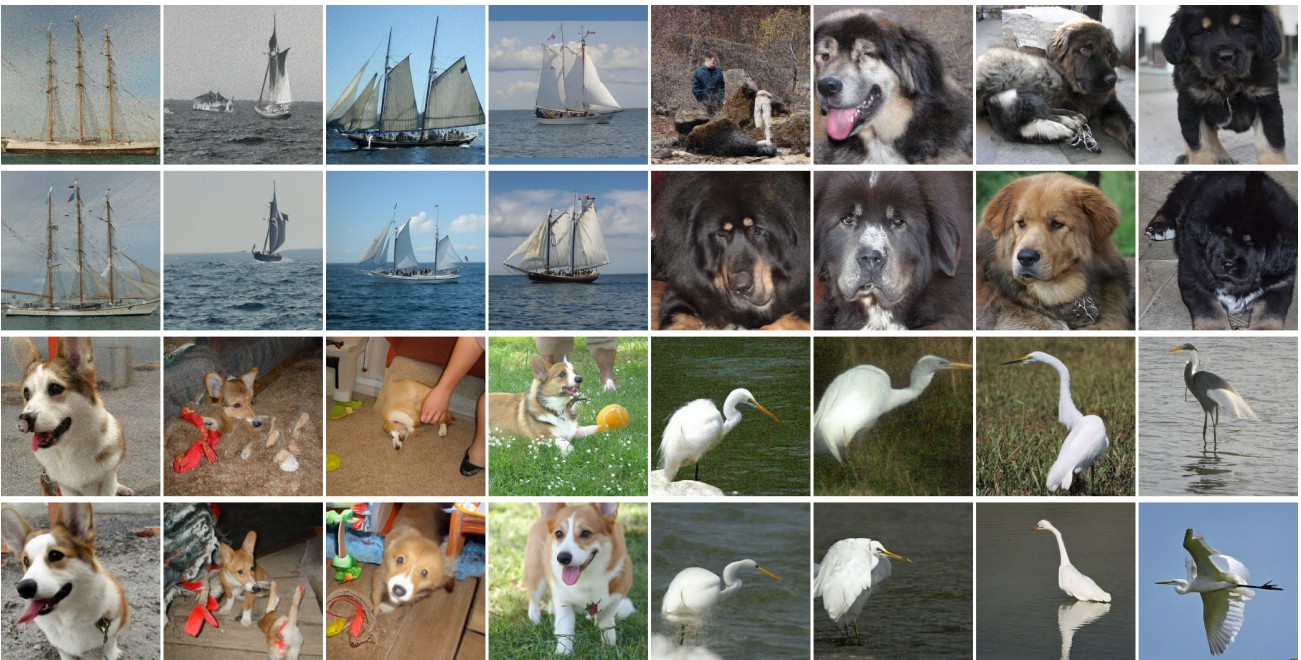

*Figure 4.* Samples from ImageNet $256 \times 256$. Each $2 \times 4$ block shows the SiD2 baseline on top and ours on bottom, while the number of denoising steps is, from left to right, 32, 64, 128, and 256. Our generations are noticeably of higher quality at low step counts.

layer, for text-to-image we apply attention pooling (Radford et al., 2021) on text embeddings followed by an MLP. There is little difference between RAPSD sampler configurations and between using the sampler or the ground truth (see Section 5.4). Training minimizes the log-likelihood via stochastic gradient descent. Now, before sampling from the diffusion model, we sample $\alpha$ and $\beta$ and proceed as usual.

Formally,

$$\{w_c, \mu_c, \sigma_c\}_{c=1}^C = S(y), \quad (24)$$

$$c' \sim \text{Cat}(w_{1:C}), \quad \{v_1, v_2\} \sim \mathcal{N}(\mu_{c'}, \text{diag}(\sigma_{c'})), \quad (25)$$

$$\beta = \exp(v_1), \quad \alpha = \frac{v_2 - v_1}{\log N_f}, \quad (26)$$

where Cat is a categorical distribution. See also Section D.2.

### 4.5. Conditioning and guidance interval

Our noise schedules require only two minor modifications to SiD2 (Hoogeboom et al., 2025) for maximum performance. The baseline conditions the denoiser on the logSNR using FiLM (Perez et al., 2018), so it is aware of the noise level. Since we have a different noise schedule for each image, more information is needed to fully determine the schedule. We additionally condition on minimum and maximum logSNR per image, making $c = (y, \lambda_M(t; x_0), \lambda_M(0; x_0), \lambda_M(1; x_0))$ in Eqs. (2) and (3).

SiD2 defined the classifier-free guidance inter-

val (Kynkäänniemi et al., 2024) in terms of logSNR, whereas, for similar reasons, we define it based on $t$. Section 5.4 shows the effect of these changes, Algorithms 1 and 2 summarize our training and sampling methods, Section D.1 provides more details.

## 5. Experiments

We experiment on class-conditional image generation on ImageNet at multiple resolutions, closely following the architecture and training protocol defined in SiD2 (Hoogeboom et al., 2025). We also report zero-shot text-to-image on MS-COCO. See Section D for implementation details.

### 5.1. Class-conditional image generation

Table 1 shows our results on class-conditional generation on ImageNet, and Figure 7 shows generated samples. We focus on comparisons with single-stage pixel diffusion models, excluding LDM and distilled models. The fairest comparison is against SiD2, which we reproduce and outperform in almost all metrics, while using fewer denoising steps, though the margin is smaller in the compute-heavy setting. Ours and SiD2 metrics are averaged over 5 runs.

While our results are strong in the single-stage pixel diffusion setting, they still do not reach the best LDMs and distilled models. As examples, RAE (Zheng et al., 2025) is an LDM that reaches 1.13 FID on ImageNet with 50 denois-

*Table 2.* Ablation studies on ImageNet $256 \times 256$ using the *small* model architecture. We analyze the impact of our main contributions, including scheduling, conditioning mechanisms, and guidance interval parametrization.

| METHOD | NFE | FID $\downarrow$ | IS $\uparrow$ |
|---|---|---|---|
| *Baselines* | | | |
| SiD2 BASELINE | 512 | 1.68 | 288.2 |
| **OURS (MIXED SCHEDULE)** | 256 | 1.42 | 297.0 |
| *Schedule Design* | | | |
| FIXED SCHEDULE (MEDIAN) | 256 | 1.61 | 299.3 |
| COSINE WITH MINMAX | 256 | 1.98 | 271.4 |
| FREQUENCY-FOCUSED ONLY | 256 | 5.16 | 178.7 |
| POWER-FOCUSED ONLY | 256 | 5.18 | 316.2 |
| *Conditioning & Sampling* | | | |
| W/O MINMAX CONDITIONING | 256 | 1.58 | 302.8 |
| LOGSNR INTERVALS | 256 | 1.58 | 287.7 |
| GT SPECTRUM (ORACLE) | 256 | 1.43 | 298.2 |

ing steps, and the distilled version of SiD2 achieves 1.50 FID on ImageNet $512 \times 512$ with only 16 denoising steps.

## 5.2. Reducing the number of denoising steps

Our "tight" noise schedules significantly outperform the baselines in the low-step regime. Figure 3 shows the FID at varying number of denoising steps. Interestingly, our noise schedules exhibit a slight worsening at high-step counts so there is an optimal count for each resolution; we investigate this further in Section C.1. Our model outperforms the baseline at the optimal count, and the gap widens at lower counts. Figure 4 compares generations.

## 5.3. Manipulating the sampled spectrum

Here we manipulate the sampled spectrum to modify properties of the generated image. After sampling $\alpha$ and $\beta$ (Section 4.4), the target RAPSD is approximated by $\tilde{\Psi}(k) = \beta k^{\alpha}$, so the power at the highest frequency $N_f$ will be $\beta N_f^{\alpha}$. Adding $\log_{N_f} c$ to $\alpha$ results in a factor $c$ being applied to the power at $N_f$, without changing the energy at the lowest frequency $k = 1$. The effect enables controlling the amount of details of the generated image. This works because our model sees a number of different spectrum-based noise schedules during training and is conditioned on their parameters. Figure 5 shows some examples. Section C.2 shows a different manipulation that controls the generated image contrast.

## 5.4. Ablations

Here we evaluate the effect of our architectural changes with respect to SiD2, as well as alternative designs that could be considered. Table 2 shows the results. Section C.4 shows extra ablations on the hyperparameters we introduce; namely, $\kappa_{\min}$, $\kappa_{\max}$, and the $t$-based guidance interval.

*Table 3.* Zero-shot text-to-image generation on MS-COCO $512 \times 512$. Our runs are based on the *flop heavy* models described in Section D.1, modified to inject text conditioning. This outperforms the SiD2 reported results even before applying our contributions, which widen the gap while needing fewer denoising steps. The ORACLE model is the same as OURS but using the ground truth RAPSDs instead of the trained sampler.

| METHOD | PARAMS | NFE | FID $\downarrow$ | sFID $\downarrow$ | IS $\uparrow$ |
|---|---|---|---|---|---|
| *Originally reported* | | | | | |
| SiD | 2B | 256 | 9.6 | - | - |
| SiD2 | - | 256 | 8.1 | - | - |
| *Our runs, heavy models* | | | | | |
| SiD2 | 647M | 128 | 7.75 | 17.3 | 33.9 |
| SiD2 | 647M | 160 | 7.56 | 16.2 | 33.6 |
| SiD2 | 647M | 256 | 7.85 | 14.9 | 33.4 |
| OURS | 649M | 80 | 7.18 | 15.2 | 34.6 |
| OURS | 649M | 96 | **7.08** | 14.9 | 34.7 |
| OURS | 649M | 128 | 7.14 | 14.4 | **34.9** |
| OURS | 649M | 160 | 7.31 | 14.2 | 34.6 |
| OURS | 649M | 256 | 7.83 | **13.9** | 34.3 |
| ORACLE | 649M | 96 | 6.55 | 14.3 | 35.0 |

**Fixed schedule (median)** Typical noise schedules are the same for all images; here we quantify the effect of varying them per-instance. We adopt the same principles from Section 4.3, but make the schedule constant by taking the median schedule over a subset of training images. While this outperforms the SiD2 baseline, it underperforms ours.

**Cosine MinMax** We evaluate the effect of using the RAPSD curves to guide the noise schedules. We use the prescriptions for minimum and maximum noise from Section 4.2 to design a per-instance schedule that follows a cosine between the extremes. It performs worse than ours.

**Frequency/Power focused** Our best noise schedule is an average of the frequency and power-focused schedules. Here we evaluate each of them independently.

**No conditioning** The only architectural modification we propose to SiD2 is the extra FiLM conditioning layers described in Section 4.5. Performance worsens without it.

**LogSNR intervals** Another change with respect to SiD2 is that we set guidance intervals in terms of $t$. Here we evaluate the usual setting in terms of logSNR, which performs worse.

**GT spectrum (oracle)** We quantify the effect of sampling the power spectrum parameters, by evaluating a model with the spectrum computed from ground truth images. Results are close, showing no loss when using the RAPSD sampler.

## 5.5. Text-to-image generation

We train text-to-image models on the same dataset as SiD2, at $512 \times 512$ resolution. The model is based on the heavy

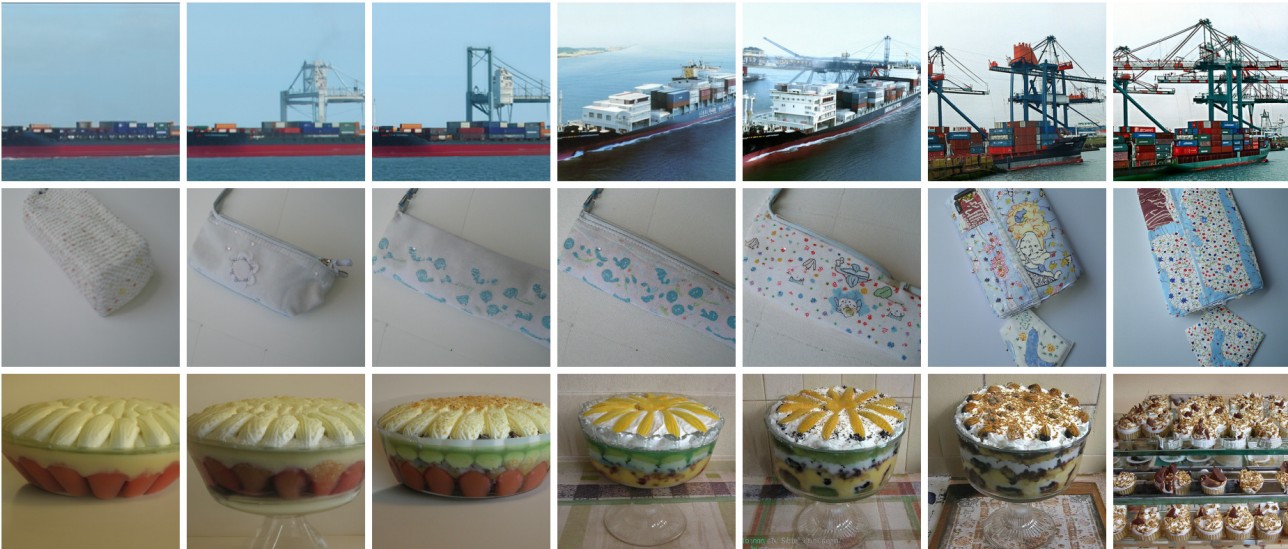

*Figure 5.* Manipulating the sampled spectrum to modify generated image properties. Here we modify the sampled spectrum such that the energy at the highest frequency is multiplied by factors 0.1, 0.2, 0.4, 1.0, 2.5, 5.0, 10.0, respectively. This affects the noise schedule and the model conditioning, so it is a way to guide the model towards different spectral properties. In this example, the energy on high frequencies correlate to the amount of texture and details. Notice how the amount of details increase from left to right. Images are generated by the same model trained on ImageNet $256 \times 256$ and same initial noise.

model used for ImageNet $512 \times 512$. The training protocol and hyperparameters are all the same, with architectural modifications to handle the text inputs. We use T5-XXL (Raffel et al., 2020) embeddings to encode the text prompts, and pass them to the model in two ways: 1) as additional inputs to FiLM conditioning after attention pooling, and 2) via cross-attention added to every transformer block.

We evaluate zero-shot text-to-image generation on MS-COCO (Lin et al., 2014), comparing against single-stage pixel diffusion models; namely, SiD (Hoogeboom et al., 2023) and SiD2 (Hoogeboom et al., 2025). SiD2 did not report implementation details about text-to-image models; our reproduction following their ImageNet *flop heavy* model outperforms their published results.

Table 3 shows the results and Fig. 8 shows generated samples. Our reproduced models already outperform the SiD2 reported numbers. When applying our spectral noise scheduling, the gap widens further, while requiring fewer denoising steps, consistent with the class-conditional task.

Interestingly, the FID degradation at higher NFE observed in class-conditional generation (Fig. 3) occurs at lower step counts in the text-to-image setting, and is also noticeable for the baseline models. Because these text-to-image models are trained on datasets much larger than ImageNet, we conjecture that stronger models are inherently more susceptible to sample degradation at excessively high step counts.

In contrast to the ImageNet results where the use of an oracle

ground-truth test set spectrum made little difference compared to the RAPSD sampler (Table 2), the text-to-image oracle model performs significantly better. We attribute this to the train and test distribution shift, since we evaluate zero-shot on MS-COCO and both the diffusion model and RAPSD sampler are trained on a separate dataset. Remarkably, the knowledge of just two parameters approximating the ground-truth RAPSD for each test image is sufficient to mitigate the train-test distribution gap.

## 6. Conclusion and limitations

This work demonstrated that more efficient diffusion noise schedules can be obtained by leveraging the image power spectrum and specializing the schedule for each instance. Our results showed improved quality over strictly single-stage pixel diffusion models, while needing fewer denoising steps, though they generally lag behind state-of-the-art latent diffusion and distilled models. We leave for future work to investigate whether similar techniques apply to these multi-stage models, noting that Skorokhodov et al. (2025) investigated the differences between latent and RGB spectra. While our noise schedules successfully adapt to different resolutions with no hyperparameter changes, other aspects of the model still need tuning; namely, the loss bias and guidance intervals. It remains to be seen whether these could also be tied to spectral properties.

## Acknowledgments

We are grateful to Emiel Hoogeboom and Leonardo Zepeda-Núñez for reading the manuscript and providing valuable feedback. We also thank the authors of Simpler Diffusion, upon which our method is built.

## Impact Statement

This paper presents work whose goal is to advance the field of Machine Learning. There are many potential societal consequences of our work, none of which we feel must be specifically highlighted here.

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

## A. Derivation of Noised RAPSD

Here we provide the derivation for Eq. (9). The noised signal is given by $z_q = \alpha_q x_0 + \sigma_q \epsilon$. By the linearity of the DFT, the frequency domain representation is:

$$\hat{z}_q(u) = \alpha_q \hat{x}_0(u) + \sigma_q \hat{\epsilon}(u). \tag{27}$$

The power spectral density (PSD) $P_{z_q}(u) = |\hat{z}_q(u)|^2$ is:

$$P_{z_q}(u) = (\alpha_q \hat{x}_0(u) + \sigma_q \hat{\epsilon}(u))\overline{(\alpha_q \hat{x}_0(u) + \sigma_q \hat{\epsilon}(u))} \tag{28}$$

$$= \alpha_q^2 |\hat{x}_0(u)|^2 + \sigma_q^2 |\hat{\epsilon}(u)|^2 + \alpha_q \sigma_q 2 \operatorname{Re}(\hat{x}_0(u)\overline{\hat{\epsilon}(u)}). \tag{29}$$

Since the noise $\epsilon$ has zero mean and is independent of the signal $x_0$, the expected cross-term vanishes: $\mathbb{E}_\epsilon[\hat{x}_0(u)\overline{\hat{\epsilon}(u)}] = 0$. Taking the expectation over the noise distribution:

$$\mathbb{E}[P_{z_q}(u)] = \alpha_q^2 P_{x_0}(u) + \sigma_q^2 \mathbb{E}[P_\epsilon(u)]. \tag{30}$$

The RAPSD $\Psi(k)$ averages the power over constant frequency magnitudes. Applying this operator to both sides, and dropping the expectation for notational brevity, we have:

$$\Psi_{z_q}(k) = \alpha_q^2 \Psi_{x_0}(k) + \sigma_q^2 \Psi_\epsilon(k). \tag{31}$$

For standard Gaussian white noise, the spectrum is flat (white), so the expected power is constant across all frequencies. Following the normalization in Eq. (6), $\Psi_\epsilon(k) = 1$. Thus, the following is valid in expectation,

$$\Psi_{z_q}(k) = \alpha_q^2 \Psi_{x_0}(k) + \sigma_q^2. \tag{32}$$

## B. Closed-form Noise Schedules

Here we derive the closed-form equations for the frequency-focused and power-focused schedules under the assumption that the RAPSD follows a power law:

$$\tilde{\Psi}_{x_0}(k) = \beta k^\alpha, \tag{33}$$

where $\beta > 0$ and $\alpha < 0$. Given the computed RAPSD $\Psi_{x_0}$, we obtain $\alpha$ and $\beta$ by minimizing the cost

$$\min_{\alpha,\beta} \sum_k \left(\log \Psi_{x_0}(k) - (\alpha \log k + \log \beta)\right)^2 \tag{34}$$

via least-squares.

Recall that the noise scaling factor is interpolated as $\kappa_t = \kappa_{\max}^t \kappa_{\min}^{1-t}$, so $\log \kappa_t = t \log \kappa_{\max} + (1-t) \log \kappa_{\min}$.

### B.1. Frequency-focused Schedule

The frequency mapping is linear: $\mu_F(t) = N_f + (1 - N_f)t$. Substituting the power law into the schedule definition (Eq. (19)):

$$\lambda_F(t) = -\log \kappa_t - \log \tilde{\Psi}_{x_0}(\mu_F(t)) \tag{35}$$

$$= -\log \kappa_t - \log(\beta(\mu_F(t))^\alpha) \tag{36}$$

$$= -\log \kappa_t - \log \beta - \alpha \log(N_f + (1 - N_f)t). \tag{37}$$

### B.2. Power-focused Schedule

For the power-focused schedule, we first compute the normalization constant $Z$ and the CDF $F(q)$.

$$Z = \int_1^{N_f} \beta u^\alpha du = \frac{\beta}{\alpha + 1}(N_f^{\alpha+1} - 1). \tag{38}$$

The CDF is given by:

$$F(q) = \frac{1}{Z} \int_1^q \beta u^\alpha du = \frac{\frac{\beta}{\alpha+1}(q^{\alpha+1}-1)}{\frac{\beta}{\alpha+1}(N_f^{\alpha+1}-1)} \tag{39}$$

$$= \frac{q^{\alpha+1}-1}{N_f^{\alpha+1}-1}. \tag{40}$$

We solve for the inverse CDF $\mu_P(t) = F^{-1}(1-t)$. Letting $y = 1 - t$:

$$y = \frac{\mu_P^{\alpha+1}-1}{N_f^{\alpha+1}-1}, \tag{41}$$

$$\mu_P^{\alpha+1} = 1 + y(N_f^{\alpha+1}-1), \tag{42}$$

$$\mu_P(t) = \left(1 + (1-t)(N_f^{\alpha+1}-1)\right)^{\frac{1}{\alpha+1}}. \tag{43}$$

Finally, we substitute $\mu_P(t)$ into Eq. (19),

$$\lambda_P(t) = -\log \kappa_t - \log(\beta \mu_P(t)^\alpha) \tag{44}$$

$$= -\log \kappa_t - \log \beta - \alpha \log \left[ \left(1 + (1-t)(N_f^{\alpha+1}-1)\right)^{\frac{1}{\alpha+1}} \right] \tag{45}$$

$$= -\log \kappa_t - \log \beta - \frac{\alpha}{\alpha+1} \log \left(1 + (1-t)(N_f^{\alpha+1}-1)\right). \tag{46}$$

## C. Additional experiments

### C.1. Study of degradation at high NFE

Figure 3 shows some FID degradation of our models at higher number of denoising steps. This effect is most evident for the heavy models at $128 \times 128$, which is also where our model fails to clearly outperform the SiD2 baseline. We conduct multiple experiments to investigate this issue; Table 4 shows all results.

First, we evaluate the baseline at up to 4096 denoising steps, and show that it also exhibits FID degradation, though milder. This suggests that the degradation is not caused by our noise schedules, but only appear earlier than the baseline because our schedules are "tight."

Second, we report all metrics for our method. Both our model and the baseline show degradation in FID and sFID and recall, while the inception score and precision do not significantly change. Moreover, there is no apparent qualitative loss of image quality. These facts support the hypothesis that the degradation is caused by a loss of diversity, not quality.

Third, we increase the amount of noise added after each backward pass during sampling to improve diversity. This is controlled by the $\gamma$ parameter which interpolates between the noise variance at $s$ and $t$ when going from step $t$ to step $s$ as described in Eq. (5). The SiD2 default is $\gamma = 0.3$; when changing it to 0.5, our results no longer exhibit degradation at high step counts, and show superior performance at 512 steps than our originally best metrics reported at 160 steps. FIDs are better than the SiD2 baseline at all NFEs, but worse than our previous except at the highest NFE. Thus, maintaining $\gamma = 0.3$ still provides the best trade-off. We found that increasing $\gamma$ for the baseline produces significantly worse results.

Finally, we reiterate our text-to-image results (Section 5.5), which show the degradation occurring at lower step counts and also noticeable for the baseline models. These models are stronger for being trained on vastly more data, which led to the conjecture that stronger models are more susceptible to high-step count degradation. This aligns with the class-conditional degradation being most apparent on $128 \times 128$ *flop heavy*, which is the combination of easiest task and heaviest model.

### C.2. Manipulating the variance/contrast

Section 5.3 shows a way to manipulate the sampled spectrum to control the amount of details in the generated image. Here we show a different manipulation that controls the contrast. By Parseval's theorem, the image variance, or contrast, is $\sum_{k=1}^{N_f} N_k \Psi(k)$. Since we approximate $\tilde{\Psi}(k) = \beta k^\alpha$, multiplying the sampled $\beta$ by a constant factor changes the variance/contrast by the same factor. Figure 6 shows the effect of applying this manipulation.

*Table 4.* Study of degradation at high NFE on ImageNet $128 \times 128$. Results show that 1) the baselines show similar but milder degradation, 2) the degradation is predominantly due to loss of diversity, not quality, and 3) increasing the noise added at each inference step by setting $\gamma = 0.5$ mitigates the degradation and brings the best results at higher step counts for our models.

| MODEL | PARAMS | NFE | FID ↓ | sFID ↓ | IS ↑ | PRECISION ↑ | RECALL ↑ |
|---|---|---|---|---|---|---|---|
| BASELINE WITH INCREASING NFE | | | | | | | |
| SID2, FLOP HEAVY | 393M | 32 | 13.3 | 14.2 | 151.2 | 0.542 | 0.519 |
| SID2, FLOP HEAVY | 393M | 64 | 6.27 | 6.21 | 193.4 | 0.662 | 0.575 |
| SID2, FLOP HEAVY | 393M | 128 | 2.96 | 4.22 | 209.9 | 0.715 | 0.619 |
| SID2, FLOP HEAVY | 393M | 256 | 1.61 | 3.71 | **215.7** | 0.727 | 0.640 |
| SID2, FLOP HEAVY | 393M | 512 | 1.30 | 3.64 | 210.9 | 0.728 | 0.646 |
| SID2, FLOP HEAVY | 393M | 1024 | 1.33 | 3.65 | 214.7 | **0.734** | 0.641 |
| SID2, FLOP HEAVY | 393M | 2048 | 1.37 | 3.71 | 214.9 | 0.731 | 0.640 |
| SID2, FLOP HEAVY | 393M | 4096 | 1.43 | 3.77 | 214.3 | 0.733 | 0.643 |
| OURS WITH INCREASING NFE | | | | | | | |
| OURS, FLOP HEAVY | 395M | 32 | 5.50 | 5.62 | 174.8 | 0.654 | 0.601 |
| OURS, FLOP HEAVY | 395M | 64 | 2.34 | 4.01 | 196.9 | 0.714 | 0.631 |
| OURS, FLOP HEAVY | 395M | 128 | 1.33 | 3.56 | 201.5 | 0.729 | **0.648** |
| OURS, FLOP HEAVY | 395M | 160 | 1.30 | **3.53** | 204.8 | 0.731 | 0.642 |
| OURS, FLOP HEAVY | 395M | 256 | 1.46 | 3.59 | 200.6 | 0.729 | 0.638 |
| OURS, FLOP HEAVY | 395M | 512 | 1.85 | 3.74 | 202.8 | 0.721 | 0.637 |
| OURS WITH INCREASING NFE, $\gamma = 0.5$ | | | | | | | |
| OURS, FLOP HEAVY | 395M | 32 | 12.9 | 10.1 | 136.5 | 0.537 | 0.562 |
| OURS, FLOP HEAVY | 395M | 64 | 5.88 | 5.52 | 178.6 | 0.654 | 0.594 |
| OURS, FLOP HEAVY | 395M | 128 | 2.99 | 4.14 | 196.1 | 0.706 | 0.626 |
| OURS, FLOP HEAVY | 395M | 256 | 1.58 | 3.64 | 203.8 | 0.726 | 0.637 |
| OURS, FLOP HEAVY | 395M | 512 | **1.28** | 3.54 | 207.7 | 0.727 | 0.645 |

*Table 5.* Unconditional generation on ImageNet $256 \times 256$. Results show the same trends as observed in the conditional models – our models achieve superior performance with fewer denoising steps.

| MODEL | PARAMS | NFE | FID ↓ | sFID ↓ | IS ↑ | PRECISION ↑ | RECALL ↑ |
|---|---|---|---|---|---|---|---|
| ADM (DHARIWAL & NICHOL, 2021) | 554M | 250 | 26.2 | 6.35 | 39.7 | **0.61** | 0.63 |
| SID2, SMALL | 397M | 512 | 17.2 | 4.95 | 60.8 | 0.55 | **0.67** |
| **OURS, SMALL** | 399M | 256 | **16.8** | **4.37** | **61.4** | 0.56 | **0.67** |

### C.3. Unconditional ImageNet

Table 5 shows results for unconditional generation on ImageNet. The task is notoriously harder than the conditional generation, but the same trends are observed – our model outperforms the fair SiD2 baseline in all metrics while needing fewer diffusion steps.

### C.4. Ablations

Here we show ablations for the new hyperparameters introduced by our method. Table 6 shows an evaluation of the minimum and maximum noise scaling factors $\kappa_{\min}$ and $\kappa_{\max}$. Table 7 shows an evaluation of the classifier-free guidance interval.

## D. Implementation details

### D.1. Architecture and training

We build on (Hoogeboom et al., 2025) and follow their architectures and training protocols. In short, the architecture is a U-ViT with initial convolutional layers, downsampling, a Vision Transformer (ViT) (Dosovitskiy et al., 2021), and mirrored for upsampling. The difference between the *small* and *flop heavy* models is solely the input patch size, where the heavy model has all feature maps and sequence lengths four times larger.

Besides the noise scheduling, we introduce two changes to SiD2 briefly described in Section 4.5:

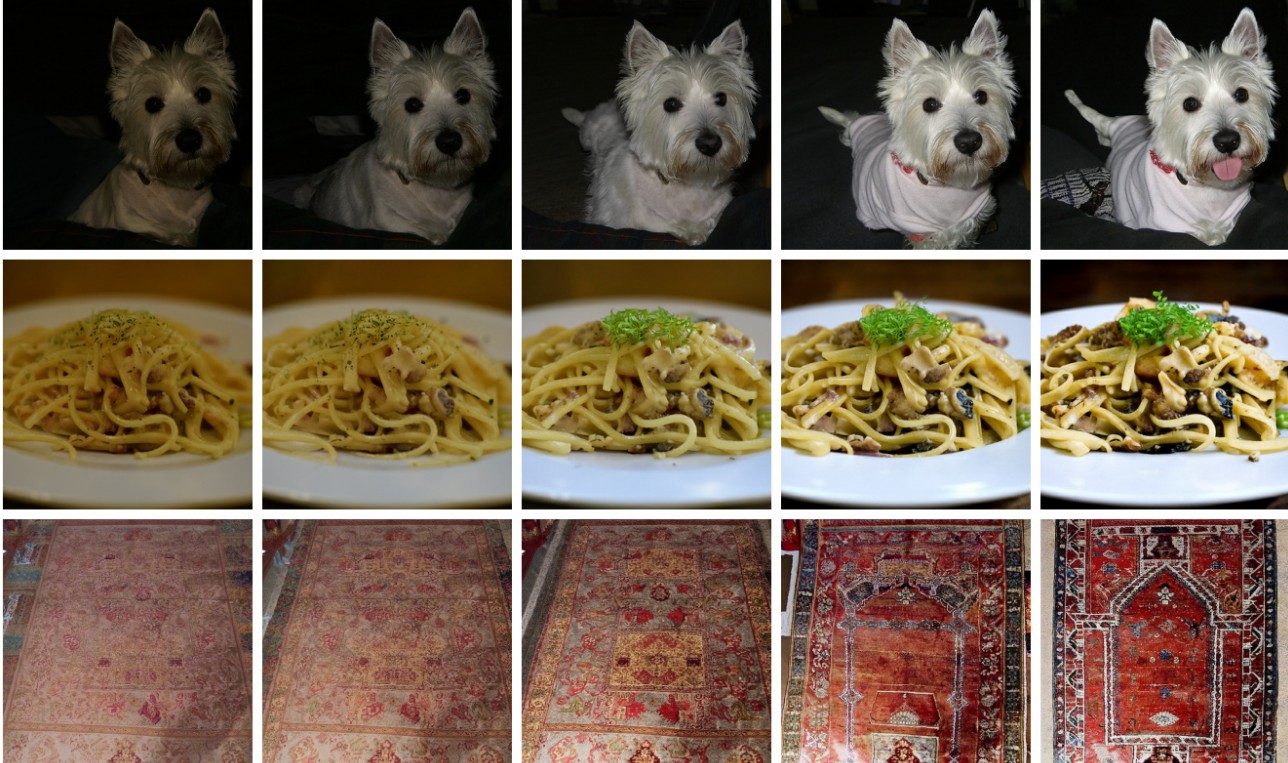

*Figure 6.* Manipulating the sampled spectrum to modify generated image properties. Here we show a different manipulation than the one in Fig. 5. We modify the sampled power spectrum by multiplying it by 1/3, 2/3, 1.0, 2.0, 3.0, respectively. The effect controls the variance of the generated pixel values (contrast). When conditioned on the modified spectra, the generated images may vary in lighting, focus, and texture. Images are generated by the same model trained on ImageNet $256 \times 256$ and same initial noise.

- FiLM (Perez et al., 2018) injects conditioning at multiple layers of the model. In the SiD2 baseline, the model is conditioned on the category/text and current logSNR. At each block, a conditioning embedding is computed from these inputs, and mapped to scale and bias parameters that are applied to the current feature map. In our approach we found it best to also condition on the shape of each schedule, so we include the min/max logSNR as additional FiLM conditioning inputs, which fully determine the schedule.

- Applying classifier-free guidance in limited intervals improves diversity and quality (Kynkäänniemi et al., 2024). SiD2 defines the interval in terms of logSNR (e.g. -1.5 to 5.0). This is not ideal for our method because we have different schedules per image. For example, for an image with maximum logSNR $< 5$, guidance would be applied all the way to the end of sampling, while a maximum logSNR $> 10$ would have many unguided steps towards the end. To resolve this issue, we define the guidance intervals in terms of the timestep $t$ instead of the logSNR.

The few hyperparameters we introduce in the diffusion model are listed in Table 8.

### D.2. RAPSD sampler

The additional model and training procedure we introduce are quite simple. We model the distribution of $\log \tilde{\Psi}_x(1)$ and $\log \tilde{\Psi}_x(N_f)$ as a mixture of Gaussians with $C$ components. For class-conditional generation, the RAPSD sampler consists of a single linear layer mapping the one-hot encoding of the class label to a vector of dimension $5C$ representing the component weight $w_c$, 2D mean $\mu_c$ and 2D diagonal covariance $\sigma_c$. For text-to-image, we apply attention pooling (Radford et al., 2021) on the text embeddings, followed by a two-layer MLP mapping to $5C$.

*Table 6.* Ablation on the scaling factors for the noise limits $\kappa_{\min}$ and $\kappa_{\max}$. Results on ImageNet $256 \times 256$, with a *small* model. We adopt $\kappa_{\min} = 0.2$ and $\kappa_{\max} = 200$.

| $\kappa_{\text{MIN}}$ | $\kappa_{\text{MAX}}$ | FID $\downarrow$ | IS $\uparrow$ |
|---|---|---|---|
| 0.2 | 200 | 1.42 | 297.0 |
| 0.2 | 100 | 1.48 | 289.8 |
| 0.1 | 200 | 1.53 | 291.0 |
| 1.0 | 100 | 1.58 | 302.9 |
| 0.1 | 100 | 1.63 | 281.0 |

*Table 7.* Effect of the $t$-based classifier-free guidance interval. Results on ImageNet $256 \times 256$, with a *flop heavy* model. We adopt (0.1, 0.45).

| GUIDANCE INTERVAL | FID $\downarrow$ | IS $\uparrow$ |
|---|---|---|
| $(0.10, 0.45)$ | 1.32 | 294.2 |
| $(0.05, 0.40)$ | 1.36 | 286.1 |
| $(0.10, 0.40)$ | 1.37 | 279.2 |
| $(0.05, 0.45)$ | 1.38 | 306.6 |
| $(0.15, 0.40)$ | 1.40 | 270.9 |
| $(0.15, 0.50)$ | 1.44 | 314.1 |
| $(0.10, 0.50)$ | 1.50 | 321.2 |
| $(0.05, 0.50)$ | 1.57 | 327.5 |

*Table 8.* Hyperparameters for different model configurations and resolutions. We list the optimal classifier-free guidance intervals (in terms of $t$), number of sampling steps (NFE), and the scaling factors $\kappa_{\min}$ and $\kappa_{\max}$.

| MODEL | RESOLUTION | GUIDANCE INTERVAL | NFE | $\kappa_{\text{MIN}}$ | $\kappa_{\text{MAX}}$ |
|---|---|---|---|---|---|
| SMALL | $128 \times 128$ | $(0.15, 0.40)$ | 160 | 0.2 | 200 |
| HEAVY | $128 \times 128$ | $(0.15, 0.35)$ | 160 | 0.2 | 200 |
| SMALL | $256 \times 256$ | $(0.10, 0.45)$ | 256 | 0.2 | 200 |
| HEAVY | $256 \times 256$ | $(0.10, 0.45)$ | 256 | 0.2 | 200 |
| SMALL | $512 \times 512$ | $(0.05, 0.50)$ | 256 | 0.2 | 200 |
| HEAVY | $512 \times 512$ | $(0.05, 0.50)$ | 320 | 0.2 | 200 |

The loss is, then, the negative log-likelihood of $v(x) = [\log \tilde{\Psi}_x(1), \log \tilde{\Psi}_x(N_f)]^\top$:

$$\mathcal{L}_{\text{GMM}}(x) = -\log \sum_{c=1}^{C} w_c \mathcal{N}(v(x); \mu_c, \operatorname{diag}(\sigma_c^2)). \tag{47}$$

The class-conditional samplers are trained for 100k steps with batch size 128, Adam with learning rate 0.01, and use $C = 5$ components. We train one sampler for each resolution and apply to all models at that resolution.

The text-conditional sampler is trained for 100k steps with batch size 128, Adam with learning rate 0.0001, and use $C = 10$ components. The feature dimension for the attention pooling and MLP is 2048.

### D.3. Metrics

We evaluate our results using standard generative modeling metrics computed on 50k generated samples.

**Fréchet Inception Distance (FID)** (Heusel et al., 2017) measures the distance between the Gaussian approximations to the distributions of Inception-V3 pool3 features of real and generated images. It is the standard metric for assessing both image quality and diversity. We measure it against the ImageNet training set.

**Spatial FID (sFID)** (Nash et al., 2021) is a variant of FID that utilizes spatial features from intermediate mixed-6/7 layers of the Inception network rather than the spatially pooled features.

**Inception Score (IS)** (Salimans et al., 2016) evaluates the distinctness and diversity of generated images based on the entropy of the predicted class distribution (Inception softmax).

**Precision and Recall** (Kynkäänniemi et al., 2019) separately assess fidelity and diversity. Precision measures the fraction of generated images whose Inception-V3 pool3 features are within the k-nearest neighbors of a real image (fidelity), while recall measures the fraction of real images that are within the k-nearest neighbors of a generated image (diversity). We use $k = 3$ and 50,000 examples from ImageNet training set for this metric.

# E. Computational cost analysis

The major difference between our model and the SiD2 baseline (Hoogeboom et al., 2025) is the RAPSD sampler. Since the sampler is small, its effect on the computational cost is negligible. We quantify the effect in this section.

Training the RAPSD Sampler on ImageNet takes around 3h on a single TPUv4 for 100k steps and batch size 128, with the job bound by data loading. For generating a batch of 512 images at 128x128 resolution with 512 denoising steps on 16 TPUv4, our model takes 71.2s while the baseline takes 71.0s. However, our best-performing models need only 160 steps instead of 512, with each batch taking only 22.3s, resulting in more than 3x speed-up. These numbers include both the RAPSD sampling and the extra dense layers that encode the RAPSD parameters conditioning as described in Section 4.5.

During diffusion model training we extract the RAPSDs from the ground truth images so the sampler is not involved. This is done in parallel during data preprocessing and does not interfere with training efficiency.

# F. Generated samples

Figure 7 shows samples generated by our *flop heavy* model trained on ImageNet $512 \times 512$. Figure 8 shows samples generated by our text-to-image model at $512 \times 512$, conditioned on MS-COCO prompts.

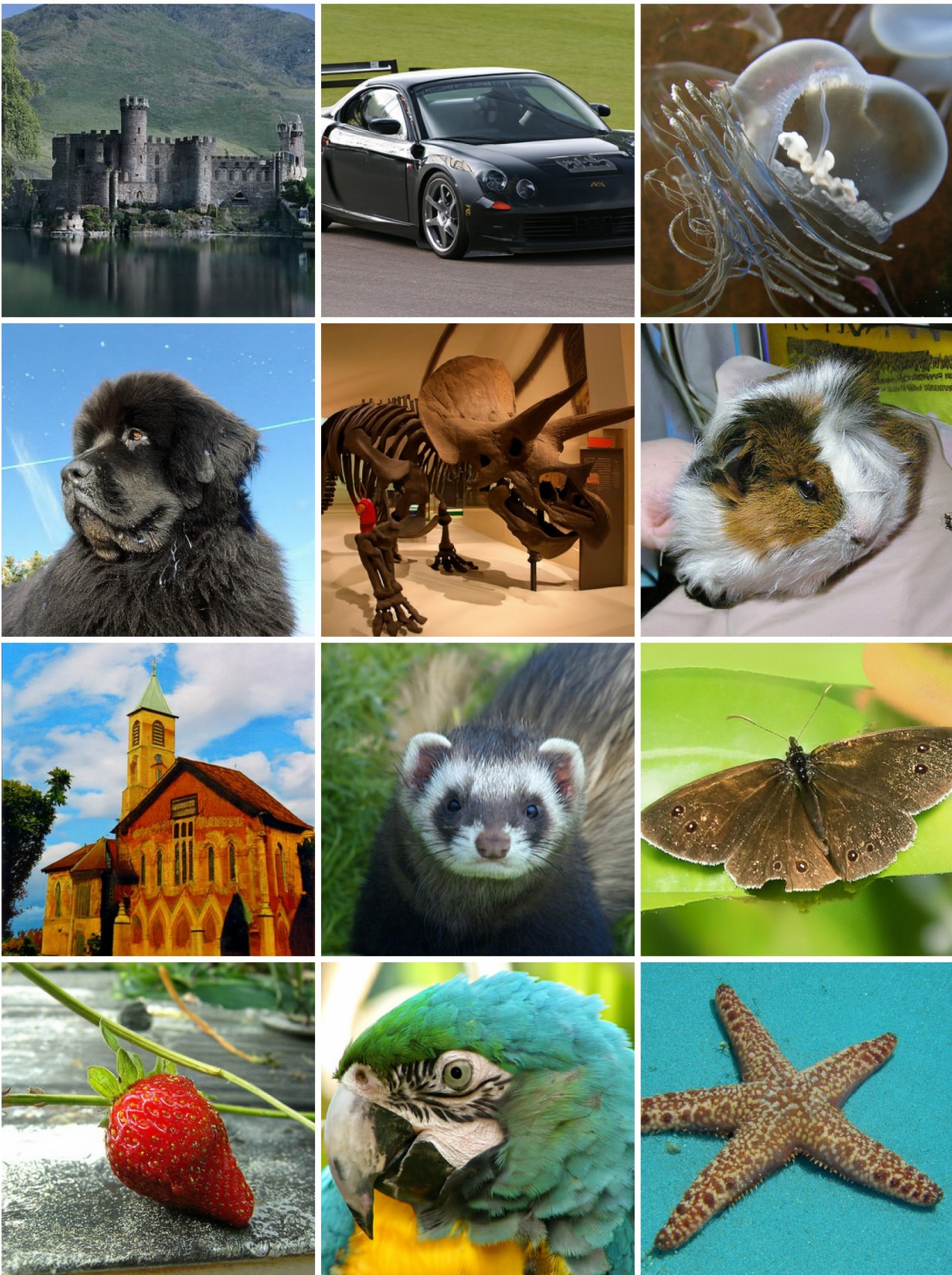

*Figure 7.* Samples generated by our class-conditional *flop heavy* model trained on ImageNet $512 \times 512$.

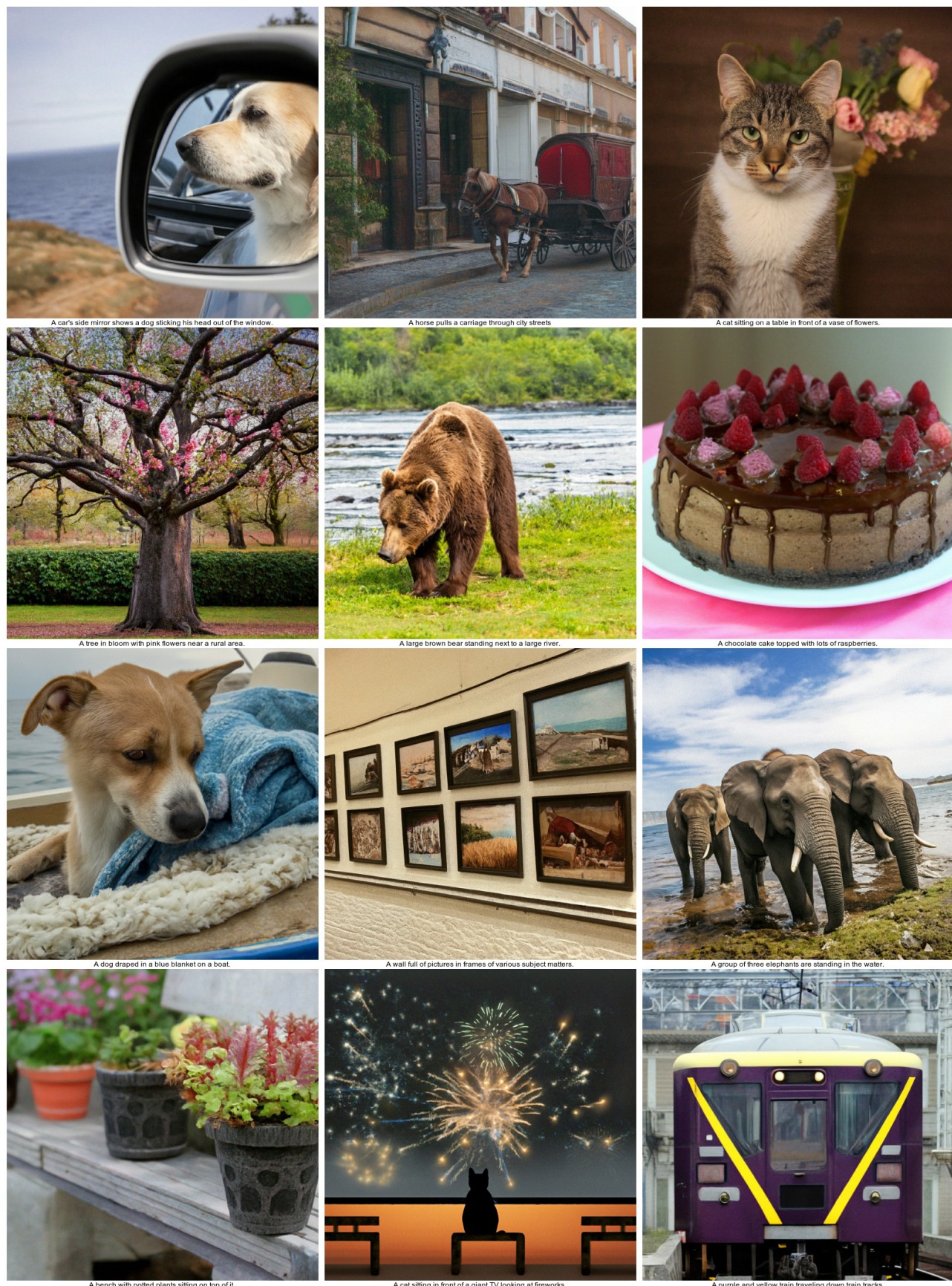

*Figure 8.* Samples generated by our text-to-image model at $512 \times 512$, zero-shot conditioned on MS-COCO prompts.

