# OpenReview forum: "Spectrally-Guided Diffusion Noise Schedules"
_ICML.cc/2026/Conference — ICML 2026 regular_

### Official Review · Reviewer_9tGL · 2026-03-12

**Soundness:** 3
**Presentation:** 3
**Significance:** 2
**Originality:** 2
**Overall Recommendation:** 3
**Confidence:** 4

**Summary:**

This paper studies noise schedule design for pixel-space diffusion models and proposes per-image logSNR schedules driven by image spectra (RAPSD).

**Compliance With Llm Reviewing Policy:**

Affirmed.

**Key Questions For Authors:**

What causes the slight quality decay at high NFE for tight schedules, and can it be mitigated?

**Limitations:**

yes

**Strengths And Weaknesses:**

Strengths:

Connects spectral structure to schedule design with an implementable per-image construction, and demonstrates strong multi-resolution ImageNet results

Weaknesses:

Requires an additional conditional mechanism (GMM-based RAPSD sampler) to predict the spectrum before sampling.

Evaluation is primarily focused on class-conditional generation, effectiveness in complex text-to-image tasks remains to be fully explored.

The method appears to slightly degrade at high NFE

---

> ### Author Rebuttal · Authors · 2026-03-31
>
> Thank you for reviewing our submission.
>
> ## W1 requires an additional conditional mechanism (GMM-based RAPSD sampler)
>
> The RAPSD sampler is a tiny model that trains quickly and has negligible computational cost during inference. In fact inference is more than 2x faster when using our method because it requires fewer denoising steps. We believe the methodological complications introduced by the RAPSD Sampler are a small price to pay for  the improved generation quality, diversity, and speed. If the reviewer is concerned with applicability to other tasks, we show below that it also works for text-to-image generation. Please refer to our [response to AJgp](https://openreview.net/forum?id=5cIgeU4WOG&noteId=rBcl27Mlda) W2, Q2 for more details.
>
> ## W2 effectiveness in complex text-to-image tasks remains to be fully explored
>
> We agree that additional experiments strengthen our claims. Following your suggestion, we trained a text-to-image model following the protocol used in ImageNet 256x256 (small), including the noise schedule and its hyperparameters.
>
> We use [T5-XXL](https://arxiv.org/abs/1910.10683) embeddings for text encoding. The only architectural change in the diffusion model is to replace category conditioning by text conditioning in the FiLM layers, which is done with a single attention pooling layer mapping the input T5-XXL sequence to a single feature. Similarly, the only change in the RAPSD Sampler is that we apply the same attention pooling + a 2-layer MLP to generate the mixture parameters. The computational cost is still negligible.
>
> We trained on a dataset of 70M pairs (image, caption) and evaluated on a held-out validation set. Results show similar trends as observed in the ImageNet experiments -- our approach brings improved quality and lower step counts.
>
> | config      | NFE | FID  | sFID | IS   | Prec. | Rec. |
> |-------------|-----|------|------|------|-------|------|
> | SiD2, small | 512 | 3.05 | 7.40 | 25.8 | 0.66  | 0.62 |
> | Ours, small | 256 | 2.43 | 7.33 | 26.4 | 0.66  | 0.63 |
>
> These results are not comparable with the ones on the SiD2 paper – we use different data and much smaller models. We will attempt to reproduce their text-to-image results and compare against their published results.
>
> ## W3, Q1 The method appears to slightly degrade at high NFE
>
> This is in fact an intriguing result of our method. Upon further investigation we observed that the qualitative image quality does not seem to decrease with higher number of steps. The metrics indicate worse FID but similar precision and IS, with clearly decreasing recall, which indicates a loss of diversity at higher NFEs. Here we report ImageNet 128x128 results, which has shown the greatest degradation:
>
> |             | NFE | FID  | sFID | IS    | Prec. | Rec.  |
> |-------------|-----|------|------|-------|-------|-------|
> | Ours, small | 32  | 5.50 | 6.00 | 182.3 | 0.663 | 0.599 |
> | Ours, small | 64  | 2.33 | 4.10 | 214.1 | 0.725 | 0.624 |
> | Ours, small | 128 | 1.45 | 3.65 | 223.2 | 0.743 | 0.635 |
> | Ours, small | 256 | 1.55 | 3.70 | 222.1 | 0.742 | 0.630 |
> | Ours, small | 512 | 1.89 | 3.83 | 220.2 | 0.745 | 0.621 |
>
> We found that increasing the amount of noise added after each backward pass during sampling helps improve diversity. This is controlled by the $\gamma$ parameter which interpolates between the noise variance at s and t when going from step t to step s. The SiD2 default is 0.3; when changing it to 0.5, we obtain:
>
> |             | NFE | FID   | sFID  | IS    | Prec. | Rec.  |
> |-------------|-----|-------|-------|-------|-------|-------|
> | Ours, small | 32  | 13.22 | 10.73 | 137.4 | 0.544 | 0.557 |
> | Ours, small | 64  | 5.93  | 5.85  | 190.1 | 0.661 | 0.585 |
> | Ours, small | 128 | 2.96  | 4.20  | 216.3 | 0.718 | 0.616 |
> | Ours, small | 256 | 1.66  | 3.65  | 224.2 | 0.738 | 0.629 |
> | Ours, small | 512 | 1.40  | 3.58  | 224.9 | 0.749 | 0.629 |
>
> Which no longer exhibits degradation at high step counts, and now shows superior performance at 512 steps than our originally best metrics reported at 160 steps. These results are better than the SiD2 baseline at all NFEs, but worse than ours with $\gamma=0.3$ except at the highest NFE. Thus, we recommend keeping $\gamma=0.3$ which provides good results with significantly fewer steps. We found that increasing $\gamma$ for the baseline produces significantly worse results. We will include a full analysis.
>
> Please also refer to our [response to kPyb](https://openreview.net/forum?id=5cIgeU4WOG&noteId=sLIf3d0GOi), W8 b).

---

> > ### Author Rebuttal · Reviewer_9tGL · 2026-04-05
> >
> > Thank you for the rebuttal. The additional experiments and analyses are helpful, particularly the new text-to-image results and the follow-up study of the high-NFE behavior. These additions strengthen the empirical case, but in my view they are not sufficient to change my overall assessment or raise my score.

---

> > > ### Author Response · Authors · 2026-04-07
> > >
> > > Thank you for replying. Since there were no follow-up questions, we
> > > provide more evidence about two weaknesses raised in the original
> > > review:
> > >
> > > ## Text-to-image experiments
> > >
> > > Here we provide results on zero-shot text-to-image generation on
> > > MS-COCO, training on a larger dataset. This is trained and evaluated at
> > > 256x256 using our small model with the text-conditioning as described in
> > > our previous
> > > [response](https://openreview.net/forum?id=5cIgeU4WOG&noteId=5NBUEA0lOr),
> > > with a total of 450M parameters. SiD2 did not report results in this
> > > setting but its predecessor [SiD](https://arxiv.org/abs/2301.11093) did,
> > > although with a model 4x larger than ours. Nevertheless, we outperform
> > > their numbers and our reproduction of SiD2 in this task. We are
> > > currently training the heavy model variants at 512x512 to compare
> > > against the numbers reported by SiD2.
> > >
> > > | config        | num params | NFE | FID | sFID | IS   | Prec. | Rec. |
> > > |---------------|------------|-----|-----|------|------|-------|------|
> > > | SiD, reported | 2B         | \-  | 8.3 | \-   | \-   | \-    | \-   |
> > > | SiD2, small   | 450M       | 512 | 8.8 | 12.4 | 31.2 | 0.56  | 0.58 |
> > > | Ours, small   | 452M       | 256 | 8.1 | 11.5 | 32.2 | 0.56  | 0.58 |
> > >
> > > Note: SiD did not report NFE and FID was the only reported metric.
> > >
> > > ## Degradation at high NFE
> > >
> > > In our second [response to kPyb](https://openreview.net/forum?id=5cIgeU4WOG&noteId=Qg0JdaJTUY) we included high NFE
> > > evaluations for the heavy models. Interestingly, the baseline model also
> > > degrades at higher number of diffusion steps, which suggests that the
> > > degradation is a property of stronger models and not necessarily caused
> > > by our contributions. We repeat the table here for convenience. Please
> > > refer to our second [response to kPyb](https://openreview.net/forum?id=5cIgeU4WOG&noteId=Qg0JdaJTUY) for more
> > > analysis, showing that increasing $\gamma$ also improves the performance
> > > of our heavy models at high NFE and outperforms our previous best
> > > results at the optimal NFE.
> > >
> > > | model       | NFE  | FID  | sFID | IS    | Prec. | Rec. |
> > > |-------------|------|------|------|-------|-------|------|
> > > | SiD2, heavy | 32   | 13.3 | 14.2 | 151.2 | 0.54  | 0.52 |
> > > | SiD2, heavy | 64   | 6.27 | 6.21 | 193.4 | 0.66  | 0.58 |
> > > | SiD2, heavy | 128  | 2.96 | 4.22 | 209.9 | 0.72  | 0.62 |
> > > | SiD2, heavy | 256  | 1.61 | 3.71 | 215.7 | 0.73  | 0.64 |
> > > | SiD2, heavy | 512  | 1.30 | 3.60 | 214.3 | 0.73  | 0.65 |
> > > | SiD2, heavy | 1024 | 1.33 | 3.65 | 214.7 | 0.73  | 0.64 |
> > > | SiD2, heavy | 2048 | 1.37 | 3.71 | 214.9 | 0.73  | 0.64 |
> > > | SiD2, heavy | 4096 | 1.43 | 3.77 | 214.3 | 0.73  | 0.64 |
> > > | Ours, heavy | 160  | 1.33 | 3.53 | 206.0 | 0.74  | 0.64 |

---

### Official Review · Reviewer_kPyb · 2026-03-12

**Soundness:** 2
**Presentation:** 1
**Significance:** 2
**Originality:** 2
**Overall Recommendation:** 4
**Confidence:** 3

**Summary:**

The paper proposed a method for setting per-pixel noise schedule in diffusion models that adapts to the Fourier spectrum of the clean image during training and replaces it with some sampled model during deployment.
Experiments show some advantages over a baseline, especially for small number of steps.

**Compliance With Llm Reviewing Policy:**

Affirmed.

**Final Justification:**

I appreciate the authors' efforts in the rebuttal and have increased my score by 1.
I expect all the additional details from our discussion to be included in the revised version. These include, among others, stating the limitations of Equations 7 and 9, clarifying the heuristic nature of the method in Sections 4.3 and 4.4 and improving the presentation, and adding the results of the additional experiments—both those appearing below and those the authors committed to.

**Key Questions For Authors:**

n/a

**Strengths And Weaknesses:**

Strengths:

1.  The motivation for per image per pixel noise scheduling is clear.

2.  Experiments show the proposed approach requires less NFEs than the baseline SID2 for reaching peak FID performance, which is slightly better than the baseline peak FID in most cases.

Weaknesses:

1.  The presentation can be improved.

2.  In Section 4.2, the derivation of Eq. 9 is presented too compactly. Add more steps to improve readability.
More importantly, the notation should make explicit that Eq. 7 is an expectation over the noise realization (utilized in Appendix A).
It does not hold exactly per realization and frequency bin. Thus, Eq. 9 is likewise valid in expectation, and the text should emphasize this.

3.  Apart from the derivation in Section 4.2 for determining min and max coefficients kappa, the proposed approach in Sections 4.3 and 4.4 is heuristic. Regarding Section 4.3, could it be that the similarity to cosine scheduling (Figure 2) is what makes your approach work as examining each of the two parts alone yield bad results in the experiments section.

4.  Section 4.4 is not clear. Details should be presented in the main paper, as replacing the spectrum of the clean x0 that is known only during training with some heuristic approximation seems to be a key ingredient of the approach.

5.  Provide more concrete details, potentially in the appendix, for the implementation issues discussed in Section 4.5.

6.  What is the computational overhead of the approach compared to the baseline?

7.  The experiments are limited to only class-conditional SiD2 model on ImageNet.
As the method is overall heuristic, and is not claimed to be model-specific, I would expect a more thorough empirical study, including also unconditional models.

8.  What can be the reason for the performance degradation of your approach when the NFE number is large? (Figure 3)
It seems that with more NFEs the baseline SiD2 can outperform your peak performance.

---

> ### Author Rebuttal · Authors · 2026-03-31
>
> We appreciate the thorough review and helpful comments. Our responses follow.
>
> ## W2: Derivation of Eq (9)
>
> You are correct, Eq (7) is valid in expectation since we used that the expected value of the noise RAPSD is 1. We will update the notation, clarify, and add the extra steps for deriving Eq (9).
>
> ## W3 a) Use of heuristics
>
> Please refer to our response to AJgp W1, Q1.
>
> ## W3 b) Similarity to cosine schedule
>
> The cosine schedule is designed to focus on intermediate noise levels, which is a property of the best known schedules (see response to AJgp). Thus, it’s no surprise that our curves resemble it. Our ablations verify that the differences wrt the cosine schedule are helpful (see Table 2). First, comparison against a fixed version of our schedule shows that different schedules per instance is better. Second, applying the min/max found in section 4.2 to the standard cosine schedule shows that following the mixed schedule is better.
>
> ## W4: Clarity of section 4.4 (Fitting and Sampling the RAPSD)
>
> We will rewrite it for clarity and add details. We leverage that the RAPSD of natural images typically follows a power law $y=x^\alpha \beta$. In log-space we have the line $\log y = \alpha \log x + \log\beta$. We use least-squares in log-space to fit $\alpha$ and $\beta$ given the ground truth.
>
> This is useful for two reasons: 1) it guarantees that the noise schedules are monotonically increasing, which is necessary for the diffusion backward process, and 2) it provides a simple parametrization of the curve using only two parameters for the RAPSD sampler.
>
> Since don't have the ground-truth RAPSD during inference, we must sample it. This is done by a separately trained RAPSD sampler, which maps the category-label to $\alpha$ and $\beta$. Even 1D Gaussian sampling using the mean/std of each parameter for each class works, but we found best to fit a mixture of Gaussians.
>
> While we could have used an expectation-maximization algorithm for fitting the GMM, for generality we chose to minimize the log-likelihood via gradient descent of a linear map to the GMM params. This allows extension to text-to-image where the linear map can be replaced by a small transformer, which is in fact what we did for the text-to-image results described in our response to 9tGL W2.
>
> ## W5: Details about conditioning and guidance intervals
>
> We will include details. FiLM injects conditioning at multiple layers of the model. In the SiD2 baseline, the model is conditioned on the category and current logSNR. At each block, a conditioning embedding is computed from these inputs, and mapped to scale and bias parameters that are applied to the current feature map. In our approach it’s best to condition on the shape of each schedule, so we include the min/max logSNR as additional FiLM conditioning inputs.
>
> SiD2 applies classifier-free guidance only within a logSNR interval (e.g. -1.5 to 5.0). This is not ideal for our method because we have different schedules per image. For example, for an image with maximum logSNR < 5, guidance would be applied all the way to the end of sampling, while a maximum logSNR > 10 would have many unguided steps towards the end. To resolve this, we define the guidance intervals in terms of the timestep t.
>
> ## W6 Computational overhead
>
> The overhead is negligible, please refer to our response to AJgp W2, Q2.
>
> ## W7 more thorough empirical study
>
> We agree that additional experiments strengthen our claims.
>
> We follow your suggestion and train our 256x256 ImageNet models and the baseline unconditionally. Results show similar trends as before, with ours outperforming the baseline with lower step count.
>
> | config      | NFE | FID   | sFID | IS   |
> |-------------|-----|-------|------|------|
> | SiD2, small | 512 | 17.25 | 4.95 | 60.8 |
> | Ours, small | 256 | 16.79 | 4.38 | 61.4 |
>
> The unconditional case is harder so the metrics for all models are worse. For reference, [ADM](https://arxiv.org/abs/2105.05233) reported an FID=26.21, sFID=6.35 and IS=39.70 on this task.
>
> Please refer to our response to 9tGL for text-to-image experiments. Notice that the only change wrt our ImageNet models is on the text-conditioning, which shows the robustness of our method.
>
> ## W8 a) Degradation at high NFE.
> Please refer to our response to 9tGL W3, Q1.
>
> ## W8 b) Does the baseline outperform at higher NFE?
>
> We found that the baseline performance saturates after some number of steps so it does not outperform our peak numbers. The table below shows evaluation of the baseline on ImageNet 256x256 with up to 4096 steps:
>
> | model       | NFE  | FID  | sFID | IS    | Prec. | Rec. |
> |-------------|------|------|------|-------|-------|------|
> | SiD2, small | 512  | 1.71 | 4.04 | 288.4 | 0.73  | 0.65 |
> | SiD2, small | 1024 | 1.66 | 3.94 | 290.0 | 0.73  | 0.65 |
> | SiD2, small | 2048 | 1.69 | 3.95 | 287.4 | 0.73  | 0.65 |
> | SiD2, small | 4096 | 1.71 | 3.95 | 286.3 | 0.72  | 0.65 |
> | Ours, small | 256  | 1.43 | 3.83 | 298.2 | 0.73  | 0.64 |

---

> > ### Author Rebuttal · Reviewer_kPyb · 2026-04-03
> >
> > I thank the authors for their response, which will be considered in my recommendation.
> > Since the approach is overall heuristic (should be emphasized in Sections 4.3 and 4.4), I think that the empirical coverage should be extended, as mentioned. Also, regarding point 8, present the results for "heavy", for which the baseline SiD2 is better.

---

> > > ### Author Response · Authors · 2026-04-07
> > >
> > > Thank you for following up. We will emphasize the heuristics aspects of
> > > our method. The following new empirical evaluations will be included in
> > > the revised version:
> > >
> > > 1.  Large scale text-to-image in settings similar to SiD2. Our latest
> > >     results show we also outperform the baselines with fewer NFE on
> > >     zero-shot text-to-image on MS-COCO. Please refer to our two
> > >     [responses to
> > >     9tGL](https://openreview.net/forum?id=5cIgeU4WOG&noteId=5VA6BAVTtX)
> > >     for details.
> > > 2.  Unconditional ImageNet.
> > > 3.  Class-conditional ImageNet at higher number of steps and different
> > >     sampler configurations.
> > >
> > > We will also include the variance/contrast manipulation experiment
> > > described in our [answer to
> > > AJgp](https://openreview.net/forum?id=5cIgeU4WOG&noteId=rBcl27Mlda).
> > >
> > > Here we provide the requested evaluations of the heavy models on
> > > ImageNet 128x128. First, we evaluate the heavy baseline at a higher
> > > number of steps. Interestingly, similar trends of degradation at higher
> > > NFE as our models now appear, though not as severe.
> > >
> > > | model       | NFE  | FID  | sFID | IS    | Prec. | Rec. |
> > > |-------------|------|------|------|-------|-------|------|
> > > | SiD2, heavy | 32   | 13.3 | 14.2 | 151.2 | 0.54  | 0.52 |
> > > | SiD2, heavy | 64   | 6.27 | 6.21 | 193.4 | 0.66  | 0.58 |
> > > | SiD2, heavy | 128  | 2.96 | 4.22 | 209.9 | 0.72  | 0.62 |
> > > | SiD2, heavy | 256  | 1.61 | 3.71 | 215.7 | 0.73  | 0.64 |
> > > | SiD2, heavy | 512  | 1.30 | 3.60 | 214.3 | 0.73  | 0.65 |
> > > | SiD2, heavy | 1024 | 1.33 | 3.65 | 214.7 | 0.73  | 0.64 |
> > > | SiD2, heavy | 2048 | 1.37 | 3.71 | 214.9 | 0.73  | 0.64 |
> > > | SiD2, heavy | 4096 | 1.43 | 3.77 | 214.3 | 0.73  | 0.64 |
> > > | Ours, heavy | 160  | 1.33 | 3.53 | 206.0 | 0.74  | 0.64 |
> > >
> > > Second, we report all metrics for our model at varying NFE. It shows
> > > similar trends as
> > > [before](https://openreview.net/forum?id=5cIgeU4WOG&noteId=5NBUEA0lOr)
> > > (similar IS and precision, but worse recall and FID), which corroborates
> > > the observation that degradation at higher NFE is predominantly due to
> > > loss of diversity, not quality.
> > >
> > > | model       | NFE | FID  | sFID | IS    | Prec. | Rec.  |
> > > |-------------|-----|------|------|-------|-------|-------|
> > > | Ours, heavy | 32  | 5.50 | 5.62 | 174.8 | 0.654 | 0.601 |
> > > | Ours, heavy | 64  | 2.34 | 4.01 | 196.9 | 0.714 | 0.631 |
> > > | Ours, heavy | 128 | 1.33 | 3.56 | 201.5 | 0.729 | 0.648 |
> > > | Ours, heavy | 256 | 1.46 | 3.59 | 200.6 | 0.729 | 0.638 |
> > > | Ours, heavy | 512 | 1.85 | 3.74 | 202.8 | 0.721 | 0.637 |
> > >
> > > Finally, we repeat the evaluation at $\gamma=0.5$ which shows similar
> > > trends as
> > > [before](https://openreview.net/forum?id=5cIgeU4WOG&noteId=5NBUEA0lOr),
> > > with our model at 512 steps outperforming our best previous results at
> > > 160 steps, while being worse at fewer NFEs.
> > >
> > > | model       | NFE | FID  | sFID | IS    | Prec. | Rec.  |
> > > |-------------|-----|------|------|-------|-------|-------|
> > > | Ours, heavy | 32  | 12.9 | 10.1 | 136.5 | 0.537 | 0.562 |
> > > | Ours, heavy | 64  | 5.88 | 5.52 | 178.6 | 0.654 | 0.594 |
> > > | Ours, heavy | 128 | 2.99 | 4.14 | 196.1 | 0.706 | 0.626 |
> > > | Ours, heavy | 256 | 1.58 | 3.64 | 203.8 | 0.726 | 0.637 |
> > > | Ours, heavy | 512 | 1.28 | 3.54 | 207.7 | 0.727 | 0.645 |

---

### Official Review · Reviewer_AJgp · 2026-03-12

**Soundness:** 3
**Presentation:** 3
**Significance:** 3
**Originality:** 3
**Overall Recommendation:** 5
**Confidence:** 4

**Summary:**

The paper considers the question of how to design noise schedules for diffusion
models. It develops an approach which allows for a systematic and theoretically founded choice of schedule, both during training and inference. Numerical
experiments demonstrate the advantages of the method, in particular in the
low-step regime. Moreover, it is shown that the methods can be used to adjust
spectral properties of the images, e.g. contrast or level of detail.

**Compliance With Llm Reviewing Policy:**

Affirmed.

**Final Justification:**

My initial positive opinion of the paper was reinforced by the authors comments in the rebuttal.
As such I remain very much in favor of accepting the paper.
This includes taking into account the other reviews. While I think they contain a number of interesting and valid points of criticism, from my point of view none of them are severe enough to prevent acceptance.

**Key Questions For Authors:**

1. Can you better motivate the choice of the mixed schedule?
2. Can you provide some information on the cost of training/using the RAPSD
sampler?
3. Can you give further information on how to use you method to manipulate
the spectrum in interesting ways akin to what was done in Section 5.3?

(While addressing these question might be helpful for future readers, it is
quite unlikely to make me increase the score to 6. Moreover, not addressing
them would not cause me to lower the score.)

**Limitations:**

yes

**Strengths And Weaknesses:**

Strengths:
1. The approach taken in the paper is well motivated, mathematically grounded,
clearly presented, and appears likely to be a useful contribution to the
topic.
2. The numerical experiments show that the adjusted noise schedule matches
(or sometimes even slightly improves on) the performance of baseline
methods while using a significantly lower number of steps.
3. Figure 5 provides an interesting showcase on how to modify the sampled
spectrum in order to guide the image generation towards certain spectral
properties. While this aspect is only briefly explored here, it could make
for interesting further work based on the presented approach.

Weaknesses:
1. The specific choice of the mixed schedule as an unweighted average between frequency- and power-focused schedule feels rather unmotivated.
2. The overhead of training resp. sampling the RAPSD sampler during training resp. inference is unclear.

---

> ### Author Rebuttal · Authors · 2026-03-31
>
> We appreciate your encouraging review and thoughtful comments.
>
> ## W1, Q1: Motivation for mixed schedule
>
> Prior work reported the benefits of sampling intermediate noise levels more often [[1](https://arxiv.org/abs/2102.09672v1), [2](https://arxiv.org/abs/2206.00364v2), [3](https://arxiv.org/abs/2407.03297v2)], since too low and too high levels during training are not sufficiently informative. We proposed the frequency and power-focused schedules as simple ways to map discrete RAPSD frequencies to continuous diffusion timesteps. It turns out that the frequency-focused results in excessive sampling of high frequencies, while the power-focused results in excessive sampling of low frequencies. The mixed schedule is then a natural way to attenuate these extremes, and the best-of-both-worlds.
>
> We acknowledge that this is a heuristic design choice. However we see it as superior to the alternatives because, besides improving generation quality and speed, the hyperparameters have a theoretical grounding and intuitive explanation, and can be the same for across resolutions and datasets. Prior work often had to tune several seemingly arbitrary numerical values. For example the baseline [SiD2](https://arxiv.org/abs/2410.19324v2) tunes logsnr_min, logsnr_max, noise_res_low, noise_res_high, while [EDM](https://arxiv.org/abs/2206.00364v2) tunes $P_{\text{mean}},P_{\text{std}},\sigma_{\text{max}},\sigma_{\text{min}},\rho$.
>
> ## W2, Q2: Cost of RAPSD Sampler
>
> Training the RAPSD Sampler is fast since the model is so small; it took around 3h on a single TPUv4 for 100k steps for batch size 128. For the same reason, the cost of the sampler during inference is negligible: For example, for generation of a batch of 512 images at 128x128 resolution using 512 denoising steps on 16 TPUv4, our model takes 71.2s while the baseline takes 71s. However our best-performing models need only 160 steps instead of 512, and each batch takes only 22.3s in this case, resulting in more than 3x speed-up. These numbers include both the RAPSD sampling and the extra dense layers that encode the RAPSD parameters conditioning as described in section 4.5.
>
> During diffusion model training we extract the RAPSDs from the ground truth images so the sampler is not involved. This is done in parallel during data preprocessing which does not interfere with training efficiency. We will include a timing analysis.
>
> ## Q3 Details on manipulating the spectrum
> We appreciate you bringing attention to this unique aspect of our method.
>
> The first step of the generation procedure is to sample $\alpha$ and $\beta$ such that the target RAPSD is approximated by $\tilde{\Psi}(k) = \beta k^\alpha$. The power at the highest frequency $N_f$ will be $\beta N_f^\alpha$. Now multiplying $\alpha$ by a factor $\log_{N_f}c$ results in a factor $c$ being applied to the power at $N_f$. In section 5.3 we use this to apply factors from 0.1 to 10 to the power at the highest frequency, which changes the whole RAPSD curve and allows controlling the amount of details in the generated images.
>
> Another way to manipulate the spectrum is applying the same factor to the power at lowest and highest frequency. This changes the contrast/variance of the image. We will add examples of such manipulation.

---

> > ### Author Rebuttal · Reviewer_AJgp · 2026-04-03
> >
> > I very much appreciate that the authors took the time to address my questions.
> > The answers were informative and further rounded out the picture.
> > I remain confident in my recommendation of acceptance.
> > While it don't believe it fully satisfies the extreme requirements for a score of 6, I would consider it to be on the upper end of 5.

---

### Decision · Program_Chairs · 2026-04-30

**Decision:**

Accept (regular)

**Comment:**

The paper presents a well-motivated and technically solid approach to noise schedule design for diffusion models, with a clear connection between image spectra and per-instance logSNR schedules.

Reviewers found the core idea interesting and practically useful, especially given the consistent gains in low-step generation and the evidence that the method can improve efficiency while maintaining or slightly improving sample quality. The rebuttal addressed several important concerns by clarifying the theoretical scope, adding implementation and overhead details, and expanding the empirical picture with additional unconditional and text-to-image results. At the same time, some concerns remain about the heuristic nature of parts of the method, the presentation clarity, and the need for broader empirical validation.

Overall, the paper makes a meaningful contribution with promising practical impact, and I lean toward a weak acceptance.